# Quantifying Spatiotemporal and Elevational Precipitation Gauge Network Uncertainty in the Canadian Rockies

André Bertoncini[1] and John W. Pomeroy[1]

[1]Centre for Hydrology, University of Saskatchewan, 200 – 50 Lincoln Park, Canmore, AB T1W 3E9, Canada

*Correspondence to*: André Bertoncini (andre.bertoncini@usask.ca)

**Abstract.** Uncertainty in estimating precipitation in mountain headwaters can be transmitted to estimates of river discharge far downstream. Quantifying and reducing this uncertainty is needed to better constrain the uncertainty of hydrological predictions in rivers with mountain headwaters. Spatial estimation of precipitation fields can be accomplished through interpolation of snowfall and rainfall observations, these are often sparse in mountains and so gauge density strongly affects

precipitation uncertainty. Elevational lapse rates also influence uncertainty as they can vary widely between events and observations are rarely at multiple proximal elevations. Therefore, the spatial, temporal, and elevational domains need to be considered to quantify precipitation gauge network uncertainty. This study aims to quantify the spatiotemporal and elevational uncertainty in the spatial precipitation interpolated from gauged networks in the snowfall-dominated, triple continental divide, Canadian Rockies headwaters of the Mackenzie, Nelson, Columbia, Fraser and Mississippi rivers of British Columbia and

Alberta, Canada and Montana, USA. A 30-year (1991-2020) daily precipitation database was created in the region and utilized to generate spatial precipitation and uncertainty fields utilizing kriging interpolation and lapse rates. Results indicate that gauge network coverage improved after the drought of 2001-2002, but it was still insufficient to decrease domain-scale uncertainty, because most gauges were deployed in valley bottoms. It was identified that deploying gauges above 2000 m will have the greatest cost-effective benefits for decreasing uncertainty in the region. High-elevation gauge deployments associated with

university research and other programs after 2005 had a widespread impact on reducing uncertainty. The greatest uncertainty in the recent period remains in the Nelson headwaters, whilst the least is in the Mississippi headwaters. These findings show that both spatiotemporal and elevational components of precipitation uncertainty need to be quantified to estimate uncertainty for use in precipitation network design in mountain headwaters. Understanding and then reducing these uncertainties through additional precipitation gauges is crucial for more reliable prediction of river discharge.

## 1 Introduction

Precipitation forcing is a primary source of uncertainty in hydrological models, therefore, accurately measuring and producing spatial estimates of precipitation is an essential step in predicting hydrological variables such as river discharge. This is especially the case in mountain headwaters where high precipitation variability is also generated by orographic enhancement.

Techniques to estimate observed precipitation use gauged precipitation networks; hence, knowing the uncertainty in these

networks is important for their optimization (Chacon-Hurtado et al., 2017) and for understanding the propagation of uncertainties in the hydrological modelling chain (Schreiner-McGraw and Ajami, 2020).

Uncertainty in precipitation estimation can profoundly affect the simulation of streamflow and other hydrological variables. Precipitation forcing uncertainty is known to degrade the quality of simulated soil moisture, evapotranspiration (Ehlers et al., 2019; Kabir et al., 2022), and ultimately, streamflow (Ehlers et al., 2019; Qi et al., 2020). In the case of streamflow, the uncertainty in precipitation can be amplified when passed down through the hydrological modelling chain (Biemans et al., 2009; Kabir et al., 2022). For instance, a 20% increase in precipitation can cause a ~ 30% increase in the annual runoff of an Arctic basin (Rasouli et al., 2014). On the other hand, a 20% decrease in precipitation can generate a ~ 40% decrease in the annual runoff of a southern boreal forest basin (He et al., 2021). Precipitation variations of such magnitude are commonly found in the uncertainty of many current precipitation products (Tang et al., 2020; Asong et al., 2017; Lespinas et al., 2015); hence, it is expected that hydrological models forced with these uncertain precipitation forcings could generate misleading streamflow predictions in preparing for drought or flooding events.

Precipitation can be measured in many ways, ranging from simple rainfall tipping buckets to shielded weighing gauges that can also measure snowfall. Most existing methods measure the amount of precipitation from a single point in space. However, precipitation is highly variable in space, and spatial fields need to be estimated to accurately represent water input to river basins (Jiang and Smith, 2003; Lehning et al., 2008; Zoccatelli et al., 2015). Several methods have been developed to spatialize precipitation from gauge observations, such as Thiessen polygons (Thiessen, 1911), Inverse Distance Weighting (IDW) (Shepard, 1968), and various kriging methods (Goovaerts, 2000). These methods all require a dense network of gauges to work efficiently. Other ways to spatially estimate precipitation include ground and satellite remote sensing (Krajewski and Smith, 2002; Hou et al., 2014; Lettenmaier et al., 2015; Skofronick-Jackson et al., 2019), Numerical Weather Prediction (NWP) outputs (Lucas-Picher et al., 2021; Milbrandt et al., 2016), and NWP reanalysis (Hersbach et al., 2020). These remote sensing and modelling techniques also have intrinsic uncertainties that can be decreased using gauge observations for calibration, assimilation, or setting initial conditions. Therefore, properly understanding precipitation gauge network uncertainty is essential to leverage each technique's strengths into an optimal precipitation estimate for hydrology.

Precipitation gauge networks are established to better represent the area for which a particular organization wants to estimate precipitation with their available resources (Chacon-Hurtado et al., 2017). These networks are often designed with a less-than-ideal gauge density or misplacement of gauges (Jing et al., 2017; Kidd et al., 2017; Daly et al., 2017) for a variety of reasons, leading to higher precipitation uncertainties in unobserved areas or elevations. Geostatistics techniques such as ordinary kriging (OK) can predict values in unobserved locations utilizing information on the variance of any physical quantity between a pair of station observations with a known distance. This variance is calculated from many station pairs to compute a semivariogram, which is the relationship between the second moment of the differences between the observed quantity at two locations and the distance between these two locations. The semivariogram is used to predict the quantity and its variance, i.e., uncertainty, at unknown locations. Hence, uncertainty rises with distance from a measuring station (Goovaerts, 1999). Other methods have been employed to interpolate climatic variables, such as precipitation, by adding auxiliary information to better inform

predictions at unobserved locations. Elevation is commonly adopted as auxiliary information, such as in the two similar techniques of kriging with an external drift (KED) (Goovaerts, 2000) and regression kriging (RK) (Hengl et al., 2007). Optimal interpolation (OI) is another method used to spatially estimate environmental variables while using physically based models as background for interpolation. OI has been used in a range of interpolation applications from groundwater information (Peli et al., 2022), snow depths (Brasnett, 1999), to precipitation in the Canadian Precipitation Analysis (CaPA) reanalysis product

(Lespinas et al., 2015). Although these techniques are useful to defining the horizontal uncertainty in precipitation, they need to include elevational uncertainty with similar importance to its horizontal counterpart.

     Uncertainty in mountain precipitation estimates is exacerbated considerably due to the introduction of spatiotemporal variability by orographic enhancement (Barros and Lettenmaier, 1994; Medina and Houze, 2003; Avanzi et al., 2021; Houze and Medina, 2005). Precipitation orographic enhancement can be produced by factors, such as upslope air flow, diurnal heating

cycles, convection generated by lee side wave motions, and different types of air mass blockage (Houze, 2012). These processes generate precipitation unevenly in a river basin, but usually precipitation increases with elevation. Orographic enhancement can be represented by prescribed lapse rates from proximal gauges measuring precipitation along an elevation profile (Thornton et al., 1997; Liston and Elder, 2006; Smith and Barstad, 2004). Precipitation lapse rates are usually higher in winter and in the front ranges of mountain regions, varying from 0.55% to 13.10% per 100 m in the French Alps (Dura et

al., 2024) and can be up to 22.08% per 100 m in the front ranges of the Canadian Rockies (Fang et al., 2019). Where these gauged elevational transects are sparse or nonexistent, the uncertainty due to lapse rate or elevation can increase (Daly et al., 2008) and in addition to horizontal uncertainty, generate greater total uncertainties. Moreover, these empirically estimated lapse rates are likely to change in the future due to the modification of atmospheric systems caused by climate change (Napoli et al., 2019; Jing et al., 2019). Atmospheric models simulate precipitation orographic enhancement by calculating moist air

lifting and hydrometeor microphysics when passing over or near an orographic barrier (Houze, 2012; Lundquist et al., 2019). Hydrological models, on the other hand, employ observed or empirical lapse rates estimated from a profile of at least two gauges and distribute precipitation forcing based on the elevation difference between the precipitation source and the spatial modelling unit (Thornton et al., 1997; Liston and Elder, 2006; Smith and Barstad, 2004).

     In the Canadian Rockies, orographic precipitation enhancement is well described through lapse rates and implemented in

atmospheric and hydrological models. Annual precipitation depths in this high mountain region can roughly double over a 1000 m elevation gain (Fang et al., 2019). However, lapse rates can vary strongly depending on the atmospheric system. For example, the June 2013 rain-on-snow event that generated unprecedented flooding in the downstream city of Calgary, Alberta had precipitation accumulations that did not vary with elevation (Pomeroy et al., 2016). Other events, especially those in spring with an easterly flow, have a higher orographic enhancement since they hold large amounts of moisture and encounter a tall

orographic barrier coming from the flat prairies (Thériault et al., 2022, 2018). The difference in uncertainty due to atmospheric systems also adds to the fact that human and transportation infrastructure varies considerably along the Canadian Rockies and this affects gauge location and investment in gauge networks. Nonetheless, this mountain range has the only North American triple continental divide between the Mackenzie, Nelson, and Columbia basins that drain into the Arctic, Atlantic, and Pacific

oceans. The so-called triple divide in Montana only drains to the Pacific and Atlantic oceans. The Canadian Rockies is also the headwaters of the Fraser and Mississippi basins. These five vast basins together account for 29% of North America's area (or 7 million km$^2$). The streamflow gauge network coverage in the Canadian portion of these river basins has also been previously shown to be suboptimal (Coulibaly et al., 2012), making the precipitation gauge network in the headwaters even more important.

Quantifying precipitation gauge network uncertainty is crucial for determining areas and elevations where gauge deployment would improve precipitation estimates. In addition, the uncertainty in mountain headwater spatial precipitation can be propagated down in the hydrological modelling chain to river discharge due to the inordinate importance of high mountain precipitation to runoff generation when compared to downstream lowlands (Viviroli et al., 2020). Current methods for estimating uncertainty in gauged precipitation focus on horizontal uncertainty and largely disregard the role of elevation, even in mountain regions where orography is a crucial form of precipitation enhancement or genesis. Therefore, it is important to use techniques capable of estimating spatial precipitation uncertainty in the three domains of space, time, and elevation. Such uncertainty estimations have yet to be performed in the world's mountain water towers such as the Canadian Rockies, where precipitation gauge deployment has been concentrated in the more accessible and densely populated valley bottoms and foothills.

The purpose of this paper is to quantify the uncertainty in gauge network spatial precipitation in the snowfall-dominated Canadian Rockies headwaters of five major river basins. The specific objectives are to (i) assess the evolution of gauge network spatiotemporal and elevational uncertainty from 1991 to 2020; (ii) analyze the impact of high-elevation gauge deployment on network spatiotemporal and elevational uncertainty; and (iii) identify gauge deployment needs in the analysed headwater river basins. To achieve these objectives, a 30-year gauge-based rainfall and snowfall database was assembled from publicly available data for a large domain of the Canadian Rockies stretching from northern Montana to Alberta and northern British Columbia (BC), and a technique that involves kriging geostatistics and lapse rates was deployed to estimate daily precipitation spatial fields and their uncertainty.

## 2 Material and Methods

### 2.1 Study Area and Period

The study area covers a large domain over the Canadian Rockies. The delimitation was defined by the Prairie ecozone boundary (E), Rogers Pass in Montana (S), the Columbia Valley Trench (W), and Pine Pass in British Columbia (N) (Fig. 1). This delimitation considered topographic features that marked the transition to lower elevations or the beginning of another mountain range, which is the case of the western limit. The south and north boundaries were defined based on regions of continuous lower elevations (i.e., passes) that make the transition to other sections of the Rocky Mountains. The south delimitation marks the transition from the U.S. Northern Rockies to the largest low-elevation gap in the Rocky Mountains.

The north delimitation is midway through a region of low-elevation peaks with similar elevations to the south delimitation. The term Canadian Rockies will be coined hereinafter as the northern part of the U.S. Northern Rockies and most of the Canadian Rockies as classified by Madole et al. (1987). The purpose of the above delimitation was to provide physiographic continuity of the analyzed mountain range regardless of political boundaries between Canada and the U.S. The study was conducted over the period between the 1991 and 2020 water years, with 30 years of analysis.

135

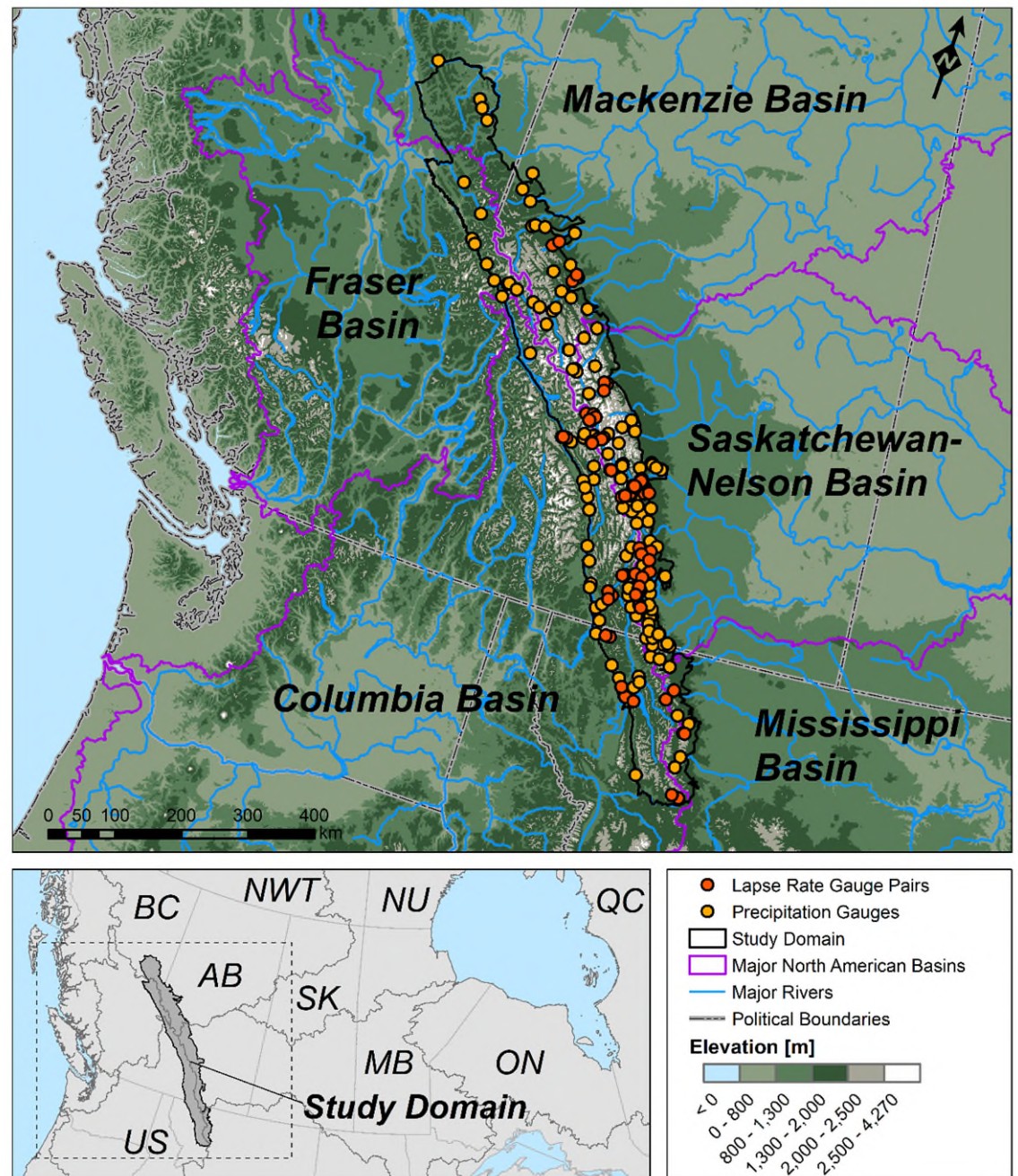

**Figure 1: Study area in the Canadian Rockies highlighting major North American headwater basins and the precipitation gauges measuring both rainfall and snowfall utilized in the analysis. Note that these gauges were not all operational at the same time. The gauge pairs shown in red were also used to define domain's daily lapse rates.**

## 2.2 Precipitation Gauge Network Inventory

An inventory was made inside the study area of precipitation gauges capable of measuring both rainfall and snowfall. Data from the following government agencies were used to compose the database (Table 1): Environment and Climate Change Canada (ECCC), Alberta Environment and Parks, Alberta Agriculture and Forestry, British Columbia Ministry of Environment, British Columbia Ministry of Transportation, US Department of Agriculture (SNOTEL), and US National Weather Service (NWS) (COOP). In addition, research gauge networks from University of Saskatchewan's Global Water Futures Observatories (GWFO) Canadian Rockies Hydrological Observatory (CRHO) and the University of Calgary were utilized. Every effort was made to search for all openly accessible precipitation gauges inside the study area. More information about each network is available in the accompanied metadata.

Table 1: Precipitation gauge networks utilized in the study with the provided time step and instrument type.

| Organization | Time step | Instrument type |
|---|---|---|
| ECCC | Daily | Alter-shielded weighing gauge |
| AB Environment and Parks | Daily | Alter-shielded and unshielded weighing gauge |
| AB Agriculture and Forestry | Daily | Alter-shielded weighing gauge |
| BC Ministry of Environment | Hourly | Standpipe |
| BC Ministry of Transportation | Hourly and 12-hour | Standpipe and manual ruler-based |
| USDA/SNOTEL | Daily | Alter-shielded weighing gauge |
| US NWS/COOP | Daily | Manual ruler-based and NWS rain gauge |
| GWFO/CRHO | 15-min | Alter-shielded weighing gauge |
| University of Calgary | 30-min and hourly | Alter-shielded weighing gauge |

Besides the traditional Alter-shielded and unshielded weighing gauges (Kochendorfer et al., 2017), this dataset is also composed of standpipe, manual ruler-based, and NWS rain gauge precipitation measurements. The BC standpipes have been shown to have a precipitation measurement precision ranging from 0.1 to 1 mm (Sha et al., 2021). They are always located in sheltered clearings where wind undercatch is minimized and considered small. For the COOP precipitation measurements, snowfall (water equivalent) can be either calculated from a manual ruler and melting the amount of snow inside a sampler or using the NWS 4 or 8-inch rain gauges and then melting the snow collected inside these gauges (NWS, 2013). COOP stations are known for having a daily negative observer bias of 1.27 mm and an observer tendency to round measurements to the nearest 0.1 or 0.05 inches (Daly et al., 2007); however, they are considered reliable stations that compose over 75% of daily precipitation stations in the US (Daly et al., 2021). In this region, they tend to be in sheltered valley bottom sites where wind redistribution is minimal. Although an important topic, it is outside the scope of this study to develop wind undercatch corrections for standpipe, ruler-based, and NWS rain gauge snowfall measurements.

## 2.3 Data Quality Control and Integration

Each network has its particular methodology to collect and quality control its precipitation and auxiliary data such as air temperature (Ta), relative humidity (RH), and wind speed (Wspd). Therefore, a methodology was developed to standardize and quality control data that was at a raw processing level. No QC was initially applied to precipitation data from ECCC, USDA/SNOTEL, US NWS/COOP, BC Ministry of Environment, and BC Ministry of Transportation. The only QC procedure applied to AB Environment and Parks and AB Agriculture and Forestry was to translate 10-m measured wind speeds to 2-m, according to Pan et al. (2017). University of Calgary precipitation data was quality-controlled following Ross et al. (2020) and Ta, RH, and Wspd based on Fang et al. (2019). All GWFO/CRHO data was quality-controlled according to Fang et al. (2019). The data was standardized by aggregating all sub-daily data into daily and ensuring that all the data was in the same time zone. Additionally, wind snowfall undercatch correction following Smith (2007) was performed because this is a region where snowfall is predominant. Partitioning between rainfall and snowfall was made using Harder and Pomeroy (2013)'s psychrometric energy budget methodology, which requires Ta, RH, and Wspd. When these variables were not available from the same organization, ERA5-Land reanalysis data at 9 km spatial resolution (Muñoz-Sabater et al., 2021) was utilized. ERA5-Land 10-m wind speed was also translated to 2-m standard measurement height following Pan et al. (2017). Surface roughness length values for wind speed translation were retrieved from Yang et al. (1998) for short vegetation and bare land, and from the Global Environmental Multiscale (GEM) model look-up-table for the remaining surfaces. European Space Agency (ESA) GlobCover 300-m landcover classification data was used for surface determination. The BC Ministry of Environment and NWS/COOP data were not corrected for undercatch since there are no existing equations for devices such as the standing pipe and manual ruler-based snowfall measurements, respectively. The different techniques and meteorological data (observed versus ERA5-Land) to correct snowfall for wind undercatch may cause inconsistencies in the precipitation dataset since it is known that ERA5-Land wind speed can be underestimated by 28 to 42% (Vanella et al., 2022); however, most snow gauge sites are in forest clearings that are sheltered from the wind, and so these inconsistencies should be smaller than the impact of not correcting the dataset for wind undercatch. Wind snowfall undercatch underestimation in the region can be up to 72% of winter monthly amounts in a high-elevation, unsheltered gauge (Pan et al., 2016). The daily precipitation data from all networks was capped at 160 mm d$^{-1}$ as a final quality assurance. The 160 mm d$^{-1}$ threshold was based on maximum daily precipitation data from ECCC's climate normals in the region. Finally, a 30-year database of daily undercatch-corrected precipitation data was composed to compute gauge network areal coverage and spatiotemporal and elevational uncertainty. ERA5-Land was chosen because it spanned the whole study period with reasonable accuracy and spatial resolution. Similar reanalysis products in the region cover only part of the study period, e.g., the Regional Deterministic Reforecast System (RDRS) covers 1980-2018 (Gasset et al., 2021) and the WRF historical run covers 2000-2015 (Li et al., 2019).

## 2.4 Precipitation Gauge Network Historical Areal Coverage

The precipitation database was utilized to compute the areal coverage of the daily gauged network. This metric quantifies the area covered by one gauge and represents areal gauge network density in km$^2$ per gauge. The network is denser (sparser) for the same unit area when the areal coverage value is smaller (larger), i.e., the areal coverage value decreases by adding new gauges. This metric is used by the World Meteorological Organization (WMO) to define the optimal number of gauges depending on environmental conditions. WMO considers 2500 km$^2$ per gauge a standard value for mountain environments (WMO, 2008). In this study, the number of daily operational gauges was employed to compute the areal coverage for the entire study domain. The areal coverage is temporally dynamic because gauges become non-operational due to missing data, seasonality or decommission, whereas they become operational due to new deployments. Elevation-segmented areal coverage was also computed by slicing the study domain into 100-m elevation bands and calculating its area and number of gauges. SRTM 90-m resolution void-filled data (Reuter et al., 2007) was used for the elevation slicing and posterior elevation data usage. Gauges that were not operational in January during the 30 years were removed from the network areal coverage analysis to alleviate seasonal signals in areal coverage.

## 2.5 Precipitation Gauge Network Spatiotemporal and Elevational Uncertainty

Precipitation gauge network spatiotemporal and elevational uncertainty was represented by the standard deviation (SD, millimetres per day) resulting from calculating daily interpolated precipitation fields by adopting a technique that merges kriging geostatistics and lapse rates. Daily precipitation gauge data ($P$) in millimetres per day was transformed to $P_Z$ [ ] using the method of Cecinati et al. (2017), to resemble a standard normal distribution with $\mu = 0$ and $\sigma = 1$ using a Normal Score Transformation (NST) before any kriging interpolation. This transformation was necessary to approximate daily precipitation that usually has a log-normal distribution skewed to zero to a Gaussian distribution, which is a requirement for kriging interpolation. Cecinati's method associates each precipitation value, in increasing order, with a value of the quantile of a standard normal distribution through a look-up-table. Repeated non-transformed values (e.g., zeros) are represented as the median of the corresponding transformed values. All the kriging interpolation and uncertainty calculations were made on the transformed data ($P_Z$), which were back-transformed at the end of the analysis for results in mm/day. Although many transformations have been commonly applied in the past for implementation simplicity (e.g., log-normal, square-root, cubic-root, and Box-Cox) (e.g., Schuurmans et al., 2007; Sideris et al., 2014; Lespinas et al., 2015; van Hyfte et al., 2023), the NST transformation resembles the Gaussian distribution the most and thus is currently used to prepare precipitation data for kriging interpolation (e.g., Cecinati et al., 2017; Lebrenz and Bárdossy, 2019). An example of the precipitation data transformation from 20 June 2019 is shown in Fig. 2. Note the gentler rise of the transformed Cumulative Distribution Function (CDF) to resemble a standard normal distribution. At the 80[th] quantile of the CDF, a precipitation value of ~ 20 mm in the non-transformed space represents a precipitation value of ~ 1 in the transformed space.

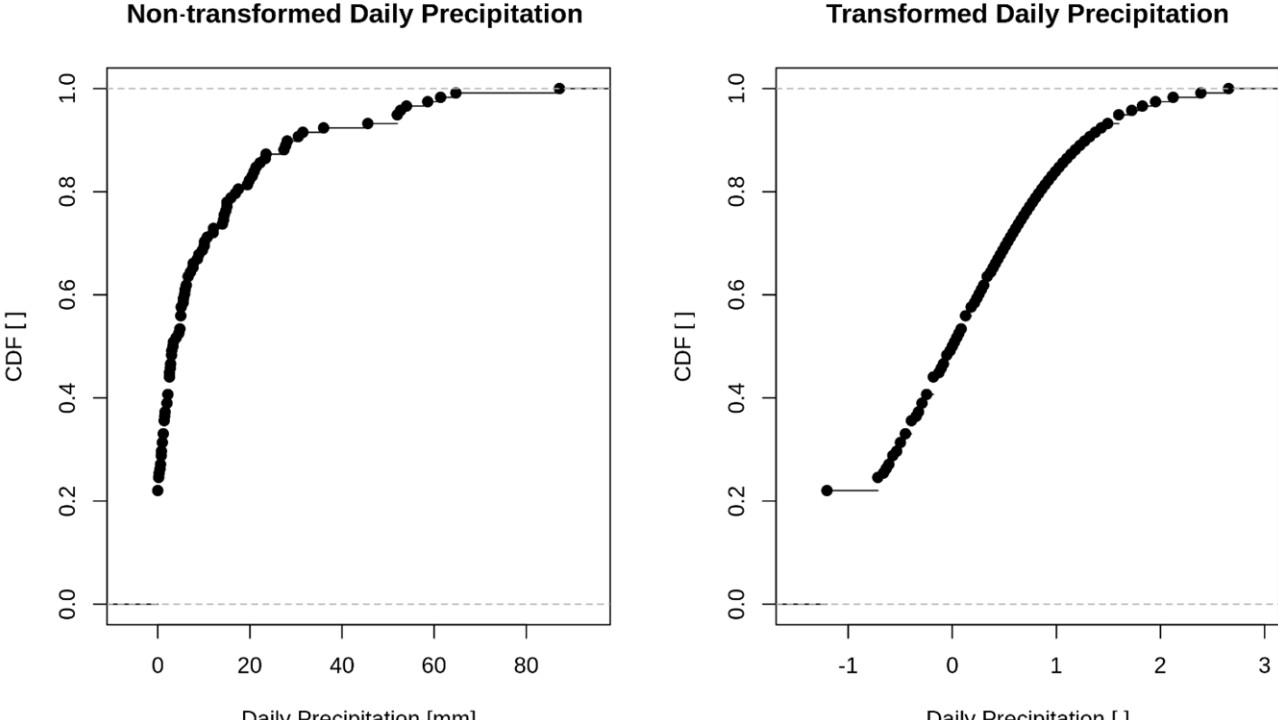

**Figure 2: Example of daily precipitation data normal score transformation for available gauges on 2019-06-20..**

Ordinary kriging interpolation on $P_Z$ was performed utilizing the *gstat* package in the R programming language (Pebesma, 2004). This package first computes a semivariogram based on latitude, longitude, and the $P_Z$ daily precipitation. The shape of the semivariogram is fitted to the data using one of the following model options: gaussian, exponential, spherical, or penta-spherical. The choice of semivariogram model options was based on the most frequent models in Ly et al. (2011), which evaluated 30 years of best fitted daily semivariogram models, and the availability in R's *gstat* package. The semivariogram model was selected based on the smallest least squares residuals between theoretical and daily precipitation sample semivariograms using the *fit.variogram* function in the *gstat* R package. An initial estimate of the sill, range, and nugget semivariogram parameters were calculated based on the shape of the sample semivariogram using the *autofitVariogram* function in the *automap* R package. The kriging technique was selected to estimate spatial precipitation mainly because it provides an estimate of interpolation uncertainty. Once the kriging interpolation was performed to the 90-m SRTM grid longitude ($i$) and latitude ($j$), the daily precipitation ($P_Z^{i,j}$) and SD ($\sigma_Z^{i,j}$) were back-transformed to $P_{mm}^{i,j}$ and $\sigma_{mm}^{i,j}$ in the units of millimetres per day. Although there are known biases associated with back-transforming precipitation and

standard deviation values, van Hyfte et al. (2023) has shown that correcting for this type of bias in a Box-Cox transformation only slightly improved precipitation estimates during summer months. The Box-Cox transformation is similar to the NST applied here (Cecinati et al., 2017). Indicator kriging, where interpolation on 0 (no precipitation occurrence) and 1 (precipitation occurrence) binary values was performed to ensure that the interpolated precipitation field did not have small lingering precipitation values where precipitation was zero. This field was calculated by inputting binary precipitation, if $P <$

0.2 mm d$^{-1}$ = 0 (trace value from ECCC) else $P = 1$, to an ordinary kriging interpolation employing the same variogram models used for precipitation magnitude interpolation. This binary 0-1 field ($P_0^{i,j}$ [ ]) was multiplied to $P_{mm}^{i,j}$ and $\sigma_{mm}^{i,j}$ to create the final daily horizontal precipitation field ($P_H^{i,j}$ [mm d$^{-1}$]) and uncertainty ($\sigma_H^{i,j}$ [mm d$^{-1}$]). All geospatial data is in the WGS84 geographic coordinate system; hence, not undergoing major differences in the distance represented by one degree of longitude in the south or north parts of the domain.

Elevational uncertainty was integrated into the spatiotemporal component to form the joint spatiotemporal and elevational uncertainty. Elevational uncertainty was calculated from daily lapse rates. The lapse rate was calculated using 53-gauge pairs located on the same hillslope with at least a 200 m elevation difference. The daily lapse rate ($\chi_\mu$) in 1 per kilometre was determined as the slope of the regression line between normalized gauge precipitation difference ($P_N$ [ ]) and elevation difference ($z_\Delta$ [km]),

$$P_N = \frac{P_h - P_l}{P_h + P_l} \tag{1}$$

$$z_\Delta = z_h - z_l \tag{2}$$


where, $P_h$ [mm d$^{-1}$] and $z_h$ [km] are the daily precipitation and elevation at the higher gauge and $P_l$ [mm d$^{-1}$] and $z_l$ [km] at the lower gauge of the same hillslope. The daily lapse rate uncertainty ($\chi_\sigma$) in 1 per kilometre was defined as the standard error of the regression line between $P_N$ and $z_\Delta$ (Thornton et al., 1997). Daily lapsed precipitation ($P_e^{i,j}$ [mm d$^{-1}$]) and lapsed uncertainty ($\sigma_e^{i,j}$ [mm d$^{-1}$]) were calculated as follows,

$$P_e^{i,j} = P_H^{i,j} \left[ \frac{1 + \chi_\mu \left( z^{i,j} - z_0^{i,j} \right)}{1 - \chi_\mu \left( z^{i,j} - z_0^{i,j} \right)} \right] \tag{3}$$

$$\sigma_e^{i,j} = \sigma_H^{i,j} \left[ \frac{1 + \chi_\sigma \left( z^{i,j} - z_0^{i,j} \right)}{1 - \chi_\sigma \left( z^{i,j} - z_0^{i,j} \right)} \right] \tag{4}$$


where $z^{i,j}$ [km] is the SRTM 90-m elevation and $z_0^{i,j}$ [km] is a reference elevation field interpolated from gauge elevations (Liston and Elder, 2006). $z_0^{i,j}$ was also generated by ordinary kriging but adopting a linear or spherical variogram model. The terms $\chi_\mu \left( z^{i,j} - z_0^{i,j} \right)$ and $\chi_\sigma \left( z^{i,j} - z_0^{i,j} \right)$ were bounded between 0 and 0.95 according to Thornton et al. (1997). When the latter terms approach 0.95 for large elevation differences, they can generate exaggerated increases in precipitation and uncertainty

due to the nonlinear nature of the Liston and Elder (2006) lapse rate implementation. To avoid these exaggerated increases, the bracketed multiplier terms in Eqs. (3) and (4) were capped at an approximate value of 8 for a ~ 2-km elevation difference, which is based on gauged lapse rate relationships in the Marmot Creek Research Basin (Fang et al., 2019). This vertical gauge profile was selected for its central placement and long precipitation lapse rate time series

The reasoning for uncertainty estimation in this study is that uncertainty in interpolated lapsed precipitation fields is not only
caused by uncertainty in spatial interpolation but also in the precipitation lapse rate. Therefore, if fewer pairs of gauges exist at high elevations or the precipitation events happening on a particular day have diverging lapse rates, the spatiotemporal and elevational uncertainty is increased. The coefficient of variation (CV) was utilized to make temporal comparisons between estimated spatiotemporal and elevational uncertainties. CV was calculated by dividing the yearly spatiotemporal and elevational uncertainty by the yearly precipitation field. Yearly fields were defined as the accumulation between 1 October
and 30 September, encompassing the northern hemisphere's water year (WY). The CV was used to represent uncertainty since using the standard deviation could mislead temporal and inter-regional comparisons. The CV is a relative measure that indicates how far the standard deviation is from the mean. A value of 1 indicates the magnitude of uncertainty is the same as the mean, and lower or higher when below or above 1, respectively. Moreover, it is common practice to use CV to indicate precipitation uncertainty resulting from kriging interpolation (e.g., Contractor et al., 2020; Phillips et al., 1992).


## 2.6 Precipitation Estimates and Uncertainty Evaluation

A leave-one-out objective verification technique was used to validate the precipitation estimates and their associated uncertainty. This technique consists of generating the precipitation estimate and uncertainty and leaving one precipitation gauge out of the analysis to be used to calculate correlation, bias, and Root Mean Square Error (RMSE) statistics. The statistics
were calculated at the daily time step and by gauge and computed for the 2000 and 2020 WYs. The distribution of precipitation errors from the leave-one-out technique was also compared to the standard deviation at unknown locations generated from the proposed method.

## 3 Results and Discussions

### 3.1 A Baseline Shift in Network Areal Coverage

A total of 206 all-weather precipitation gauges were found in the study area inventoried during the 30 years analyzed. It is worth mentioning that these 206 gauges were not all operational simultaneously, and only 163 gauges were operational year-round to compute the network areal coverage. Figure 3 shows how the historical change in the number of gauges influenced gauge network areal coverage. A clear shift in domain areal coverage due to an increase in the number of gauges was observed
around 2003. This shift occurred after the 2001-2002 drought (Bonsal and Regier, 2007; Wheaton et al., 2008), which fostered

the deployment of many gauges, especially in the Canadian Rockies' eastern foothills due to investment in monitoring by the Government of Alberta and the establishment of the Canadian Rockies Hydrological Observatory (CRHO) by the University of Saskatchewan's Centre for Hydrology. Another reason pertains to the automation of many precipitation gauges from ECCC and the Government of Alberta, which allowed year-round gauge operation in remote locations. The timing is consistent with

the decrease in manual ECCC stations around the turn of the century (Mekis et al., 2018). Before this major drought event, the domain areal coverage was sometimes greater than the 2500 km$^2$ per gauge WMO recommendation for mountain regions on a regular seasonal basis with the cessation of operation of many gauges in winter (WMO, 2008). The increase in gauging in 2003 and 2004 improved the domain areal coverage considerably which dropped to ~ 1500 km$^2$ per gauge. Spikes in areal coverage occurred because of short non-operational periods in the gauge networks.

Figure 3 also illustrates the improvement of areal coverage in high elevations. Most gauges were below 1500 m before 2003, a typical valley bottom elevation in the region. After that, many gauges were deployed to 2200 m with the establishment of Marmot Creek Research Basin as part of CRHO by the University of Saskatchewan Centre for Hydrology. Since 2013, gauge deployment at higher elevations of up to 2500 m is due to the expansion of the CRHO to Fortress Mountain, Peyto Glacier, Athabasca Glacier, and Burstall Creek, now operated as part of the national Global Water Futures Observatories observational

facility. The latter shows that even a few gauges installed at high elevations can cause a large enhancement in network coverage because of the relatively small areas at high elevations.

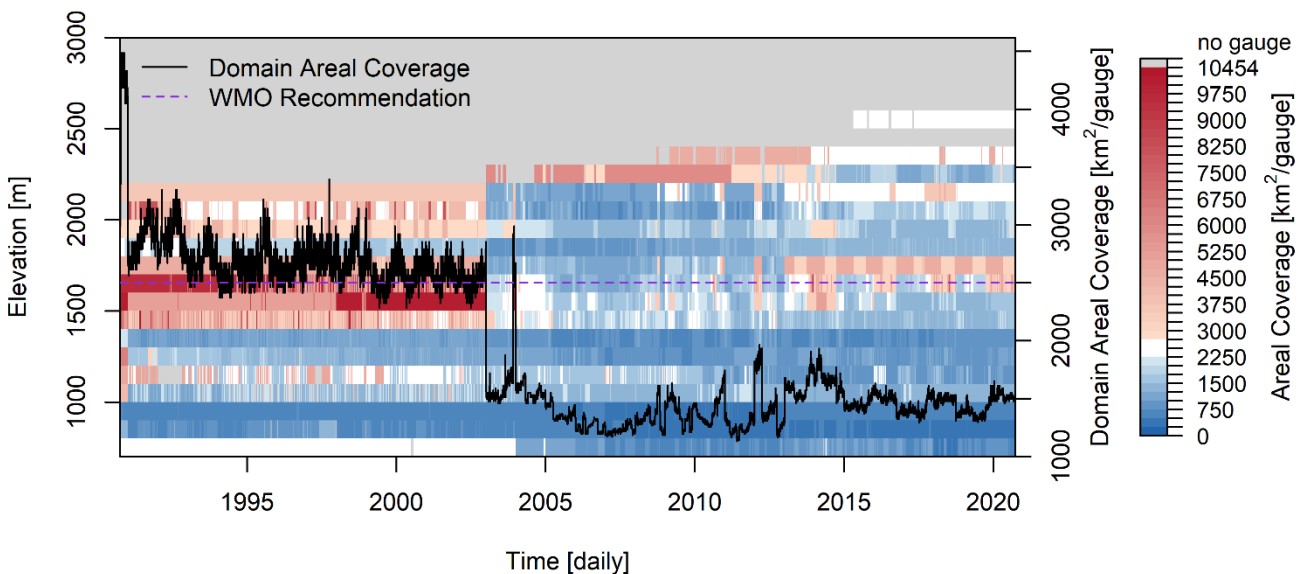

**Figure 3: Daily gauge network areal coverage per 100 m elevation bands. White represents areal coverage between 2250 and 2750**
**km$^2$ per gauge encompassing the limit of the WMO recommendation for mountain regions of 2500 km$^2$ per gauge or less. Grey denotes elevations and days with no gauge coverage. Note that a lower areal coverage value means better precipitation monitoring.**

**3.2 Precipitation Estimates and Uncertainty Evaluation**

The kriging method utilized to estimate horizontal precipitation and its associated uncertainty had a semivariogram model fit
frequency as follows for 2000: 38% for gaussian, 26% for spherical, 23% for penta-spherical, 12% for exponential, and 0.82%
with not fit. Precipitation was set to zero when the latter happened. The frequency of semivariogram model fit selection for
2020 was as follows: 35% for gaussian, 26% for exponential, 21% for spherical, and 19% for penta-spherical. The
semivariogram selection frequency found here was slightly higher than those found by Ly et al. (2011), since they had a higher
number of models to select from. Gaussian semivariograms were also the most selected in Ly's study, with the main change
being the low frequency of exponential in 2000 and a high frequency of penta-spherical models in the two years of this study.
Figure 4 illustrates the leave-one-out validation of precipitation estimates per precipitation gauge, i.e., each boxplot point
represents the daily statistics for one year of data for one gauge. Correlations were ~ 0.6 for both years (2000 = 0.64 and 2020
= 0.56), bias were -0.42 mm/day in 2000 and -0.69 mm/day in 2020, and RMSE were 3.73 mm/day in 2000 and 4.58 mm/day
in 2020. For comparison, correlations found here outperform IMERG satellite-based and CaPA modelling/assimilation
precipitation estimates, which are usually below 0.5 in the Montane Cordillera of western Canada (Asong et al., 2017). The
red triangles in the bias boxplot represent the plus and minus mean standard deviation at unknown locations computed from
the proposed methodology, indicating that most of the precipitation estimation errors are well within the estimated standard
deviation.

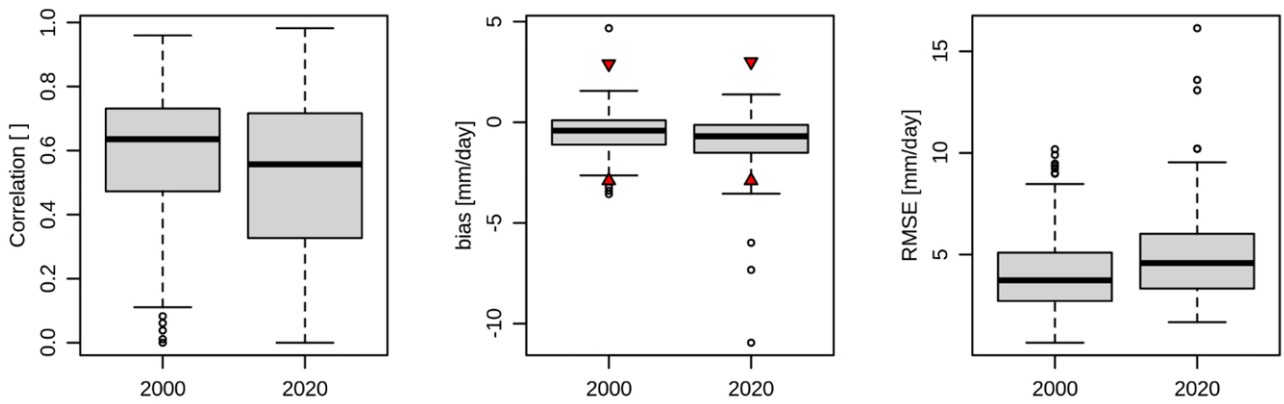


**Figure 4: Leave-one-out validation statistics distribution for the 2000 and 2020 WYs. Each boxplot point represents the daily statistic
for one year of data for one gauge. The red triangles in the bias boxplot represent the plus and minus mean standard deviation at
unknown locations computed from the proposed methodology**

Because geographic coordinates were used in this study to calculate the variograms and perform kriging interpolation, a separate leave-one-out validation using great-circle distances in kilometres for the 2020 WY was conducted to rule out any major anisotropic effects that this choice could have introduced in the results. The leave-one-out validation using kilometric distances presented the following mean statistics: correlation = 0.52, bias = -1.00 mm/day, and RMSE = 4.96 mm/day. These statistics reveal a slight degradation of using distances in kilometres over using them in degrees. In addition, there were

difficulties of fitting the reference elevation surface $z_0^{i,j}$ for one particular high-elevation gauge (Fisera Ridge at 2325 m), which required the use of additional variogram models. Still, $z_0^{i,j}$ was not fitted for nine days of the 2020 WY for this particular gauge and precipitation was set to zero. Therefore, it is concluded that using distances in degrees did not have any major influences on the precipitation estimates for the conditions of this study and the alternative approach introduced uncertainties into the analysis. Future studies should assess whether degrees or kilometric distances are the better choice for their domain

conditions.

### 3.3 Network Spatiotemporal and Elevational Uncertainty Evolution

    Daily spatiotemporal and elevational uncertainty was aggregated yearly for two WYs of particular interest for a better understating of annual accumulated uncertainties. One WY before the 2001-2002 drought (2000) and the most recently

analyzed WY (2020). Figure 5 displays the annual accumulated precipitation (a) alongside the annual accumulated standard deviation (b) for 2020. In 2020, uncertainty was small in the valley bottoms and large at high elevations. Uncertainty was larger in northern high elevations, especially around the study domain highest's peak – Mount Robson. Figure 5c illustrates the CV difference (CVΔ) from 2020 minus 2000. Uncertainty fell in Montana (CVΔ ~ -0.5) and in and north of Jasper National Park (CVΔ ~ -1.0 to -0.5) due to denser gauge deployment in less rugged terrain; the Kananaskis Valley region (CVΔ ~ -1.3

to -0.5) due to a high density of gauge deployment; and other isolated pockets (CVΔ < -0.5) due to gauge deployment in deep valleys. However, uncertainty rose in the upper Bow River basin and in and south of Kootenay National Park (maximum CVΔ > 2.5), and around Mount Robson (CVΔ ~ 1.0 to 2.5). The latter uncertainty rises were caused by gauge relocation in very complex terrain, and possibly due to differences in the spatiotemporal variability of events that happened in 2020, which can pose challenges to less dense sections of the network to capture. Some gauge deployments in deep valleys did not generate an

improvement in uncertainty, which explains why even with a higher number of gauges in 2020, a decrease in the spatial coverage of uncertainty was not widespread in the study domain. In the extreme north, there were insufficient gauges to perform kriging interpolation in 2000, which prevented the calculation of CV differences, but it may be surmised that uncertainty declined here due to gauge installation in previously ungauged regions. Note that as shown in Figure 1, lapse rates might have been underrepresented in the north part of the domain due to a lack of valid gauge pairs on the same hillslope.


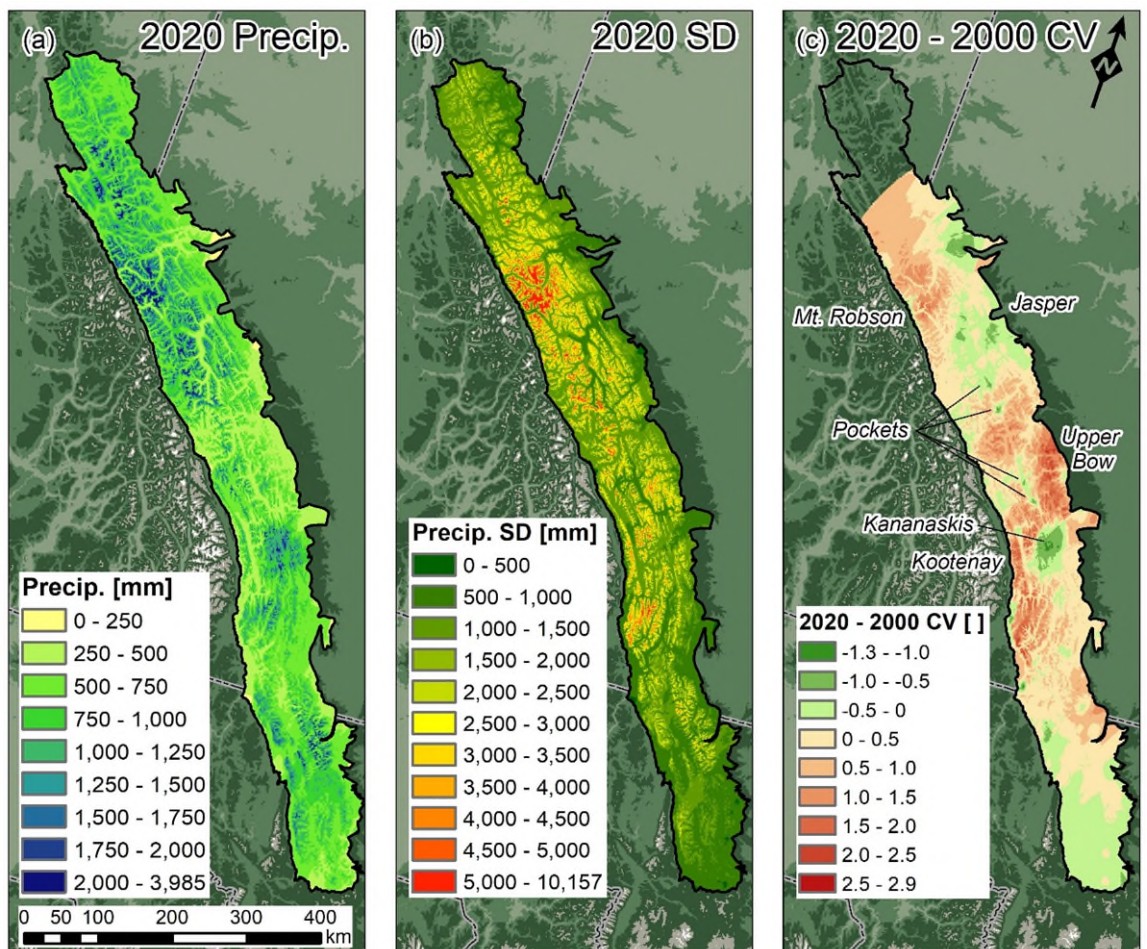

**Figure 5: Annual accumulated precipitation (a), precipitation standard deviation (SD) (b), and annual CV difference from 2020 minus 2000 (c). Note that for (a) and (b), the colour palette is stopped near the 99% quantile for better visualization.**

Precipitation uncertainty in the most recent years was higher in high elevations than in the valley bottoms. Surprisingly, although the WMO (2008) coverage recommendation range for mountain regions was reached after 2003, the overall domain uncertainty did not decrease as much as expected. This calls into question the value of increasing the density of gauge locations at lower elevations. The Storms and Precipitation Across the Continental Divide Experiment (SPADE) that took place in the southern Canadian Rockies observed that 11 out of 13 spring storm events had 30-600% higher precipitation at the well-

instrumented Fortress Mountain sites (~2000-2500 m) when compared to a gauge that was only ~ 5 km distant and ~ 500 m lower in elevation (Thériault et al., 2022). The physiographic conditions in SPADE were similar to those found in the isolated pockets of decreasing uncertainty shown in Fig. 5c, which indicates that installing further gauges at low elevations such as valley bottoms cannot represent precipitation sufficiently well when interpolated to high elevations, even when considering

lapse rates. Uncertainty is exacerbated when lapse rates vary substantially between storms and differ notably from the widely
employed values found in Thornton et al. (1997).

Although several studies have utilized kriging interpolation to assess spatiotemporal precipitation uncertainty (Goovaerts, 2000; Kyriakidis et al., 2001; Cai et al., 2019; Lebrenz and Bárdossy, 2019; Masson and Frei, 2014; Chacon-Hurtado et al., 2017), to the authors' knowledge no study has addressed the addition of elevational uncertainty with the proper contribution to overall precipitation estimation uncertainty in mountain regions. Kriging options that take elevation as a secondary variable,
such as KED (Goovaerts, 2000) and RK (Hengl et al., 2007), only provide accurate results for time steps that are longer than daily because they require moderate to high correlation between precipitation and elevation. In the hereby study, a daily time step was necessary to account for different atmospheric systems' varying lapse rates, which have shown to be highly variable in the region (Thériault et al., 2022; Pomeroy et al., 2016) and likely elsewhere as well. By implicitly accounting for lapse rate uncertainty in the kriging implementation, this study's method advances upon KED and RK that rely on regression coefficients
of precipitation and elevation relationships. These coefficients assume that these relationships are unbiased; hence, disregarding a large proportion of precipitation estimation uncertainty in mountain regions. This mechanism might be the reason KED and RK only work at moderate to high precipitation vs. elevation correlation coefficients. The resulting advantage of the novelty implemented in this work is that by accounting for lapse rate uncertainty, the uncertainty estimation is closely related to dynamic real-world scenarios in which precipitation may or may not increase with elevation.


## 3.4 The Impact of High-elevation Gauge Deployments

Mountain regions provide a unique opportunity to reduce precipitation uncertainty by deploying a few new gauges in critical high elevation areas. Although precipitation network uncertainty increases with elevation, the area in each elevation band decreases. A relatively small area of ridges and peaks need only to be covered by a few gauges. Figure 6 shows that the
elevation band area increases up to 1500 m of elevation with a gentle increase in uncertainty. Above 1500 m, the elevation band area decreases and the uncertainty increases abruptly until ~ 3000 m. At these high elevations uncertainty is the highest, but elevation band area is small. This characteristic provides an opportunity to decrease uncertainty in the studied domain by strategically placing gauges in elevations above 2000 m, where the required coverage area starts decreasing more abruptly. Despite the highest uncertainty being present in elevations above 3000 m, there are logistical challenges to install and maintain
gauges at these wind-exposed, high alpine elevations and the wind-induced gauge undercatch creates an additional uncertainty. Fortunately, they represent a small area of the study domain but they are often the accumulation zones of glaciers and so have importance for characterizing the mountain cryosphere.

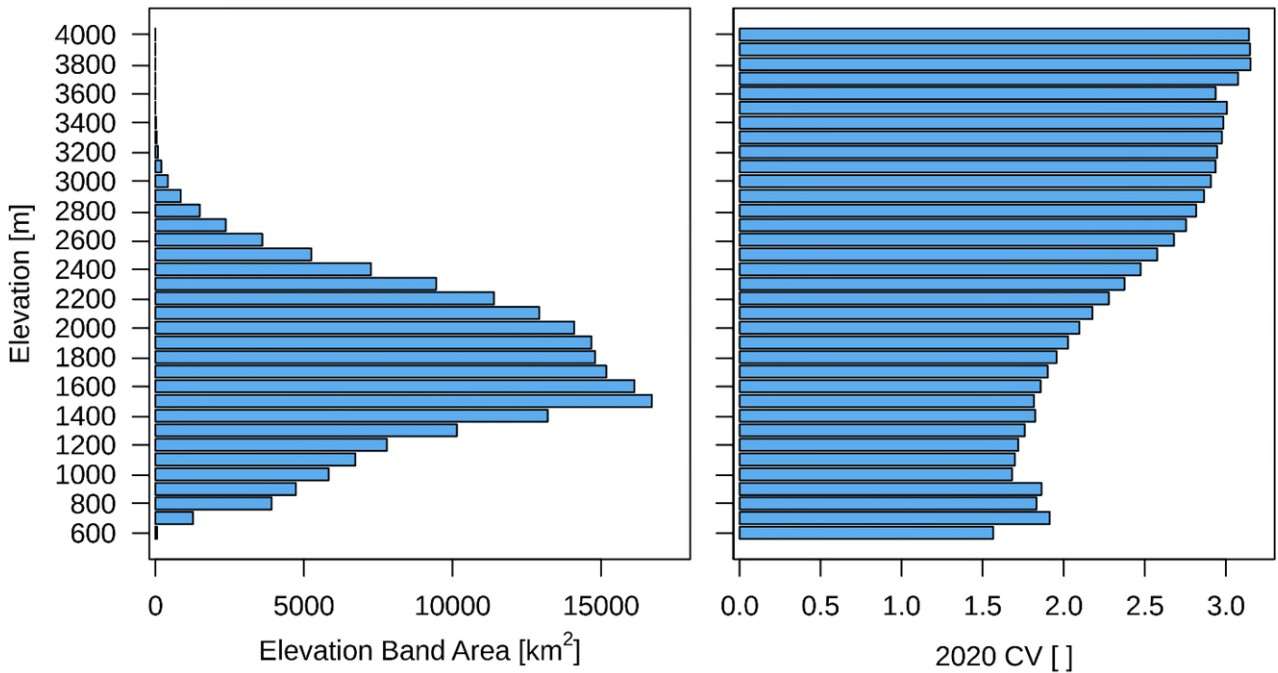

 **Figure 6: Elevation band area [km²] (left) and the mean 2020 CV [ ] (right) in the x-axes for each 100 m elevation band and their band elevation [m]. The y-axis elevation is the upper limit of the elevation band.**

The characteristic observed in Fig. 6 was corroborated in Fig. 7. The latter figure exhibits a zoomed-in map of Fig. 5c in the Kananaskis Valley region, west of Calgary, Alberta. This study area is known for gauges deployed at high elevations as part

of the CRHO of the Global Water Futures Observatories project. There is a noticeable hotspot of network uncertainty decrease in this section. This hotspot was caused by the deployment of five gauges above 2000 m of elevation in the Marmot Creek, Burstall Creek, and Fortress Mountain research basins. The deployment of these gauges decreased the local network CV up to ~ 1.3 while also maintaining a widespread impact of approximately 50 km radius in the nearby ridges and peaks. Not only in this study have high-elevation gauge deployments been shown to greatly decrease uncertainty in spatial precipitation. Brunet

and Milbrandt (2023) have demonstrated that optimally designed networks usually favour the placement of new gauges in mountain regions of Alberta and British Columbia, Canada. Brunet and Milbrandt's study suggests that, in some cases, the placement of two to three gauges can have a very significant impact on reducing network uncertainty. For instance, improving network areal coverage from ~ 1600 to ~ 1300 km²/gauge can decrease current network uncertainty close to an optimally designed network created from a blank slate in Alberta. The results shown in Fig. 7 reveal the potential that high-elevation

gauge deployment has on decreasing precipitation uncertainty estimated from gauge networks in mountain regions.

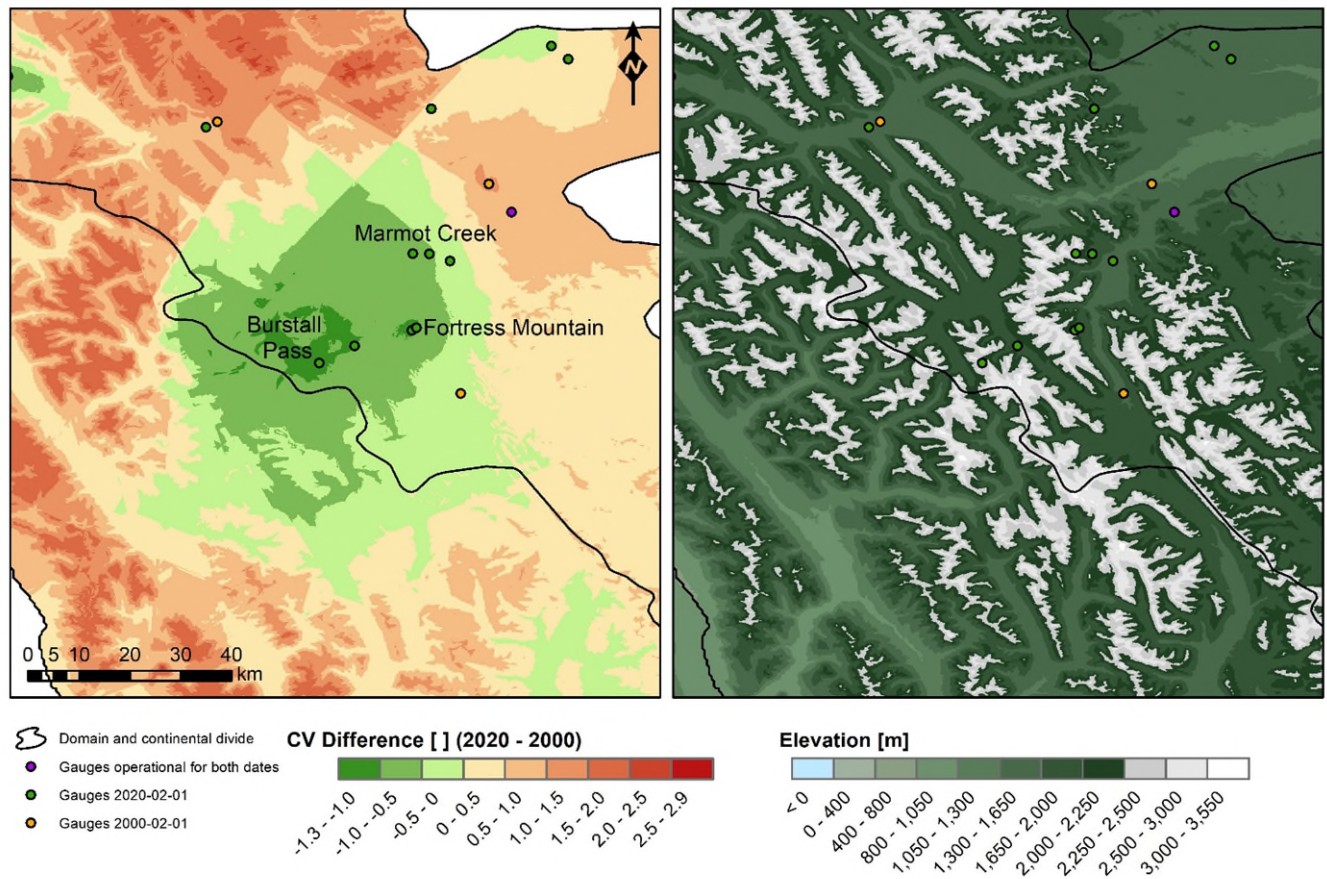

**Figure 7: Annual CV difference from 2020 minus 2000 (left) and elevation map (right) in the Kananaskis region, west of Calgary. Both maps show the gauges that were present on the first day of February 2000 and the ones deployed up to the same day in 2020, as well as gauges that were operational on both dates.**

### 3.5 Gauge Deployment Needs in the Major North American Headwater Basins

The relative need for gauge deployment in the major North American headwater basins of the Canadian Rockies is given in Fig. 8. The gauge deployment need was assumed to be proportionate to the uncertainty in precipitation as indexed by the CV. The 2020 CV 50 and 90% quantiles were 1.88 and 2.93, respectively. The need for gauge deployment should be seen as highest for CV values around and above 3, as these are in the upper end of the CV values inside the study domain. The basin with the highest uncertainty is the Nelson ($CV_\mu = 2.37$), with large CV variability between its northern, central, and southern sections. The northern part of this headwater (upper Bow River basin) is composed of high elevation mountains that are not sufficiently covered by gauges – most gauges are in the valley bottoms. These high elevations have the highest need for gauge deployment. The central part of the Nelson headwater was discussed in Sect. 3.4. This part presents the most well-distributed gauges in a

wide range of elevations, resulting in relatively low need for further gauge deployment. The southern part of this headwater is relatively well covered by gauges in mid to lower, but not at higher elevations. The headwater basin with the lowest uncertainty is the Mississippi ($CV_\mu = 1.26$), which is characterized by relatively lower elevations and a small area, simplifying gauge coverage by the network. The Mackenzie, Fraser, and Columbia headwaters had similar uncertainties of 1.93, 1.97, and 2.02,

respectively; however, there is considerable variability in the uncertainty within each basin. The central part of the Mackenzie ($CVmax = 3.61$) and Fraser ($CV \sim 3.50$) basins have the greatest need for gauge deployment. The Fraser CVmax of 6.43 happened in a limited area in its northern part, which also demands attention, but it can most likely be solved by placing one strategically located gauge. The region stretching from the Kootenay National Park and the US border has the greatest need for gauge deployment in the Columbia basin ($CVmax = 4.32$).


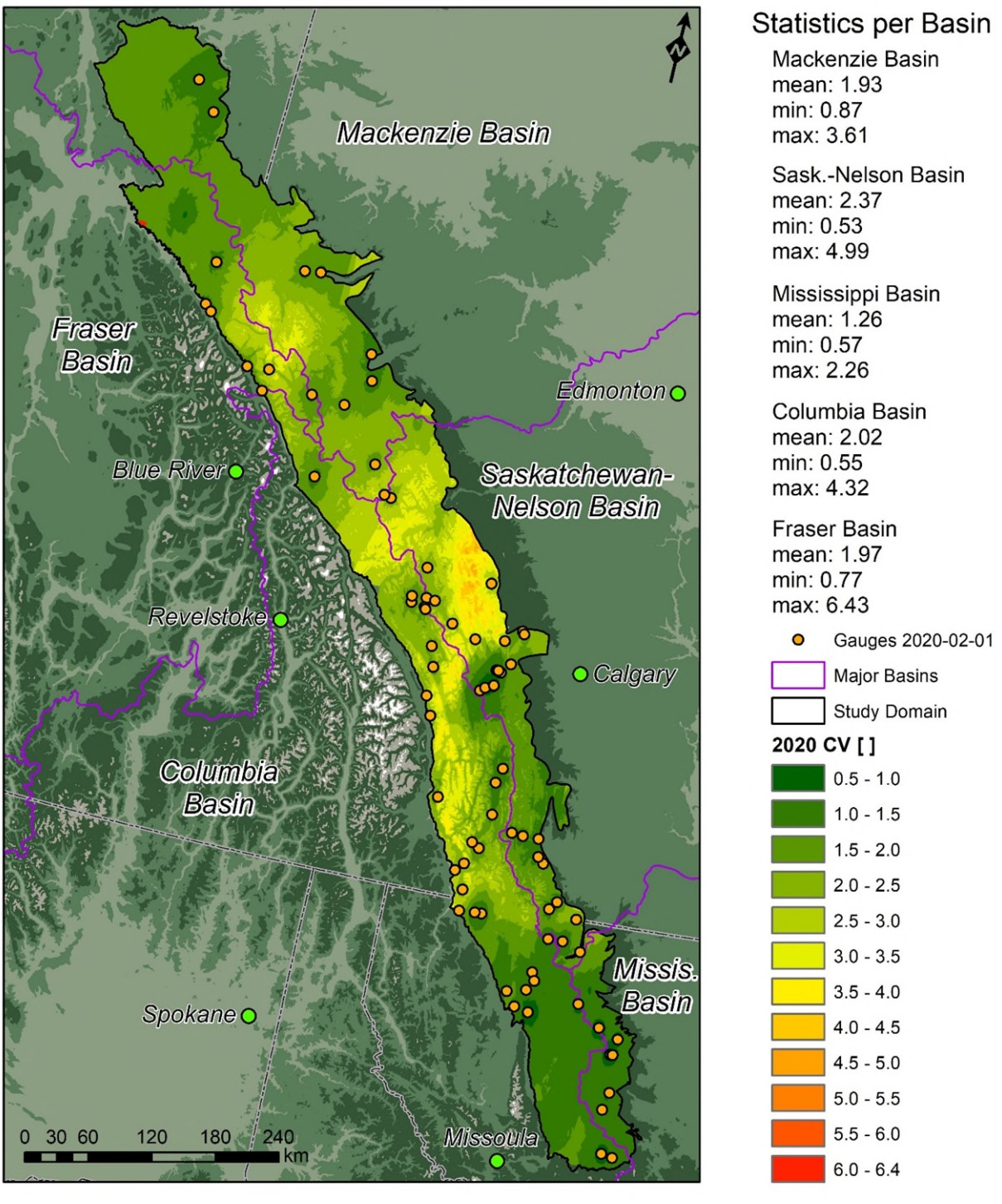

**Figure 8: Annual 2020 CV map highlighting the major North American headwater basins and the gauges operational on the first day of February 2020. Zonal statistics per basin are also displayed.**

The uncertainty described in this section suggests that precipitation in some Canadian Rockies headwater basins is still under-observed, which adds uncertainty to calculating the water resources of these basins. Even after the Calgary 2013 flood (Pomeroy et al., 2016), aside from research efforts, precipitation monitoring coverage did not improve considerably in the Nelson-Saskatchewan headwater that drains into Calgary. This finding points out a major monitoring gap that could have helped events like the Calgary 2013 flood to be better forecasted (Pomeroy et al., 2016). Better monitoring in the upper Bow River basin could have helped not only to quantify and model important storms characteristics such as precipitation amounts, extent, and duration (Milrad et al., 2017; Li et al., 2017), but also to determine the spatial distribution of the portion of this storm that fell as snow on snow-free ground. This storm snowfall component melted rapidly due to ground heat flux, contributing to 1/5 of the water input to flooding (Pomeroy et al., 2016). In addition, another major monitoring gap is found in the Columbia River headwaters, which is concerning considering that previous studies have projected an increase in the magnitude and frequency of precipitation-driven high flows due to climate change (Queen et al., 2021; Chegwidden et al., 2020). The findings demonstrated in this section reinforce the crucial need for real-time precipitation monitoring for streamflow forecasting and prediction in these major North American headwater basins.

## 4 Conclusions

This research quantified the precipitation gauge network spatiotemporal and elevational uncertainty between the 1991 and 2020 WYs in a large domain of the snowfall-dominated Canadian Rockies. The study found that precipitation network areal coverage drastically improved after the 2001-2002 drought and more recently over higher elevations due to the development of the university-operated Canadian Rockies Hydrological Observatory of the Global Water Futures Observatories national facility. Although the number of gauges has increased drastically, the deployment of many gauges in low elevations and valley bottoms did not have a widespread impact on the spatiotemporal and elevational precipitation uncertainty over large areas of the domain. Precipitation spatiotemporal and elevational uncertainty decreases and increases between 2000 and 2020 WYs occurred with similar coverage inside the domain, with the largest improvement and worsening found in the Kananaskis and upper Bow River basin regions, respectively. The findings suggest that new gauge deployments at elevations above 2000 m will have the greatest impact in decreasing the uncertainty while requiring the least number of gauges due to the decrease in coverage area at high elevations. The impact of such high-elevation gauge deployments can decrease precipitation spatiotemporal and elevational uncertainty as expressed with the CV by ~ 1.3 with a widespread (~ 50 km radius) influence on nearby ridges and peaks. The Nelson basin is the most under-observed headwater basin of the Canadian Rockies ($CV_\mu$ = 2.37), with its large uncertainty driven by the low number of gauges in the high-elevation upper Bow River basin. This suggests that the Upper Bow River Basin has the greatest need for gauge deployment, and that this should be at high elevations. The Mississippi headwaters had the lowest recent uncertainty with a $CV_\mu$ = 1.26. These findings show that the increase in gauge density in the analysed network was enough to collectively bring the Canadian Rockies to comply with the WMO

recommendation for mountain regions; however, local uncertainties remain relatively large in many high-elevation and remote areas.

The methodology developed in this study was able to quantify a mountain region's precipitation network elevational uncertainty with equitable importance to its spatiotemporal counterpart. Previous studies that utilized kriging to assess precipitation gauge network uncertainty in mountain regions have only included elevation as secondary information, and hence elevational uncertainty was largely disregarded. This study applied a technique that explicitly includes lapse rate uncertainty into the kriging implementation at a daily time step, which allows for the uncertainty caused by varying lapse rates of differing atmospheric systems to be accounted for. This advancement has major implications for assessing and reducing the uncertainty of mountain precipitation estimates since lapse rates can vary considerably from event to event and are likely to be less stable in a changing climate. By identifying areas of higher precipitation estimation uncertainty and highlighting the importance of deploying high-elevation gauges, this study offers a path forward in resolving inaccuracies in hydrological modelling through the optimization of the existing and optimal design of new precipitation gauge networks in the headwaters of major North American river basins and other cold mountain regions. Moreover, quantifiable precipitation uncertainty is crucial for the determination of uncertainty propagation in the hydrological modelling chain. Ultimately, defining the uncertainty in precipitation can help water managers to use hydrological predictions in a more informed fashion for decision-making in moments of water-related extreme events such as floods, droughts, and wildfires.

**Code Availability**

Code to generate the daily precipitation fields and accompanied spatiotemporal and elevational uncertainty is available at (https://github.com/andrebertoncini/precip_uncertainty).

**Data Availability**

Undercatch corrected daily precipitation data (with gauge metadata) used to generate the daily precipitation fields and accompanied spatiotemporal and elevational uncertainty is available at (https://github.com/andrebertoncini/precip_uncertainty). The remaining data will be made available upon e-mail request to the corresponding author.

**Author Contribution**

André Bertoncini (AB) developed the framework to generate the daily precipitation fields and accompanied spatiotemporal and elevational uncertainty. AB also performed further analysis using the precipitation gauge spatiotemporal and elevational uncertainty estimated in the Canadian Rockies, including assessing the impact of high-elevation gauges on network uncertainty and gauge deployment needs. AB and John W. Pomeroy (JWP) contributed to the hypothesis, framework, instrumentation, manuscript conceptualization, writing, and editing.

**Competing Interests**

The authors declare that they have no conflict of interest.

**Acknowledgements**

We wish to thank the European Space Agency (ESA), NASA, Google Earth Engine (GEE), and the organizations listed in Table 1 for providing this study's data and cloud computing. We also thank Xing Fang for quality controlling the GWFO precipitation and meteorological data. This work was supported by the Canada Research Chairs programme; Natural Sciences and Engineering Research Council of Canada (NSERC); Alberta Innovates; the Canada First Research Excellence Fund's (CFREF) Global Water Futures programme, and the Canada Foundation for Innovation's Global Water Futures Observatories project and various equipment grants. This research was enabled in part by support provided by the Digital Research Alliance of Canada (https://alliancecan.ca/en).

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
