# Peer review of "Quantifying Spatiotemporal and Elevational Precipitation Gauge Network Uncertainty in the Canadian Rockies"

_EGUsphere, 2024_

## Referee Comment (RC1)

**Manuscript Title:** Quantifying Spatiotemporal and Elevational Precipitation Gauge Network Uncertainty in the Canadian Rockies

**General review:** Bertoncini and Pomeroy quantify uncertainty in precipitation estimates using a network of in-situ precipitation gauges in the triple continental divide area of the Canadian Rockies, a region where precipitation can vary immensely across elevation bands and differently, depending on storm systems. Using the WMO guidelines for station density in mountainous areas, the authors transform and back-transform precipitation data (for normalized distribution), quantify a cumulative distribution function, and use a kriging and lapse rate approach to calculate and track precipitation standard deviation and coefficient of variation across space and time. In this way, the authors are able to determine areas where precipitation estimates are more and less uncertain and the areas where added in-situ observations would be most valuable in the future (e.g., relatively higher elevations). Much of the analysis, and thus manuscript text, includes a very clear description of the methodology used. I include no major changes to the workflow and thank the authors for their thorough depiction of the work in text and figures and for making the important and relevant connection to downstream hydrology. Limitations to the work, including the select reanalysis product, could be further discussed. As could a connection to other spatially distributed precipitation products. Otherwise, the small number of minor suggested changes I have made are with respect to clarifying language around some of the statistics (e.g., when uncertainty "rose" vs. "fell") and domain description. The following line-by-line comments should provide more clarity with these items, with the goal of better emphasizing the importance and value of this work, which I envision will serve as a frequent reference for many future projects.

**Line-by-line comments**
Line 121: Please define the threshold used to determine "continuous lower elevations."

Line 157: Please describe why the ERA5-Land reanalysis product was selected over other similar products.

Line 186-188: Please provide a rationale for implementing this transformation and then later back-transforming with respect to the need for a normalized distribution prior to kriging.

Figure 2: Suggest, within the associated text, interpreting the two panels for readership. I.e., what does a daily precipitation value of 60 mm versus 2 (unitless) mean with respect to CDF? Also please specify where these example data came from.

Line 232: Please specify why the lapse rate relationships (and associated rate caps) in the Marmot Creek Research Basin have been selected here.

Line 280-281: Please list average change values associated with each area when mentioning, "Uncertainty fell in Montana" and "uncertainty rose in the upper Bow River basin", etc. It would also be helpful to read additional details on these areas – similar to the way Mount Robson was listed as the study domain's highest peak.

Like 345: Please define "very significant."

Line 411-414: Within the previous discussion section, suggest the authors point to identified limitations to the methods and/or provide a clear rationale. The series of steps taken within the methodology is well cited, but could be better defended – e.g., why ordinary and indicator kriging over other methods used in the introduction? The answer may be because these methods are frequently used – not just in past studies but in ongoing snowpack and hydrologic modeling research and operational applications. On that note, it would be impactful if the authors linked this approach to modeled or satellite-based precipitation products, which likely use some of these in-situ observations and cover these domains.

---

## Author Comment (AC1)

**Quantifying Spatiotemporal and Elevational Precipitation Gauge Network Uncertainty in the Canadian Rockies**

**Bertoncini and Pomeroy (2024)**

**HESS Manuscript EGUSPHERE-2024-288**

**Author Comment 1**

**\*Note that authors responses are in blue.**

**RC1 Comments**

**Manuscript Title:** Quantifying Spatiotemporal and Elevational Precipitation Gauge Network Uncertainty in the Canadian Rockies

**General review:** Bertoncini and Pomeroy quantify uncertainty in precipitation estimates using a network of in-situ precipitation gauges in the triple continental divide area of the Canadian Rockies, a region where precipitation can vary immensely across elevation bands and differently, depending on storm systems. Using the WMO guidelines for station density in mountainous areas, the authors transform and back-transform precipitation data (for normalized distribution), quantify a cumulative distribution function, and use a kriging and lapse rate approach to calculate and track precipitation standard deviation and coefficient of variation across space and time. In this way, the authors are able to determine areas where precipitation estimates are more and less uncertain and the areas where added in-situ observations would be most valuable in the future (e.g., relatively higher elevations). Much of the analysis, and thus manuscript text, includes a very clear description of the methodology used. I include no major changes to the workflow and thank the authors for their thorough depiction of the work in text and figures and for making the important and relevant connection to downstream hydrology. Limitations to the work, including the select reanalysis product, could be further discussed. As could a connection to other spatially distributed precipitation products. Otherwise, the small number of minor suggested changes I have made are with respect to clarifying language around some of the statistics (e.g., when uncertainty "rose" vs. "fell") and domain description. The following line-by-line comments should provide more clarity with these items, with the goal of better emphasizing the importance and value of this work, which I envision will serve as a frequent reference for many future projects.

Thank you for these general comments.

**Line-by-line comments**

Line 121: Please define the threshold used to determine "continuous lower elevations."

Thank you for the suggestion. The thresholds for continuous lower elevations were based on passes that make the transition to other sections of the Rocky Mountains. More details about these passes were given in Lines 121-123. The indicated sentence was also changed to "The south and north boundaries were defined based on regions of continuous lower elevations (i.e., passes) that make the transition to other sections of the Rocky Mountains." for clarity.

Line 157: Please describe why the ERA5-Land reanalysis product was selected over other similar products.

Thank you for the comment. ERA5-Land reanalysis product was used because it had all the variables necessary for snowfall undercatch correction at a reasonable, not optimal, spatial resolution and accuracy while spanning the whole study period between 1991 and 2020. Other products exist for the region, but they only cover part of the study period. For instance, the Regional Deterministic Reforecast System (RDRS) only spans between 1980 and 2018 (Gasset et al., 2021), and the WRF historical run in the region only covers between 2000 and 2015 (Li et al., 2019). The following sentence was added in Line 167 to better describe the choice of ERA5-Land: "ERA5-Land was chosen because it spanned the whole study period with reasonable accuracy and spatial resolution. Similar reanalysis products in the region only cover part of the study period, e.g., the Regional Deterministic Reforecast System (RDRS) covers 1980-2018 (Gasset et al., 2021) and the WRF historical run covers 2000-2015 (Li et al., 2019)."

Line 186-188: Please provide a rationale for implementing this transformation and then later back-transforming with respect to the need for a normalized distribution prior to kriging.

Thanks for the suggestion. A distribution that resembles a normal distribution is a requirement for kriging interpolation. The sentence in Lines 188-189 was modified to "This transformation was necessary to approximate daily precipitation that usually has a log-normal distribution skewed to zero to a Gaussian distribution, which is a requirement for kriging interpolation." as an explanation of the need for transformations. The back-transformation is just to see the results in mm/day again. The sentence in Lines 193-194 was changed to "All the kriging interpolation and uncertainty calculations are performed in the transformed data ($P_Z$), which are back-transformed at the end of the analysis for results in mm/day." for clarity.

Figure 2: Suggest, within the associated text, interpreting the two panels for readership. I.e., what does a daily precipitation value of 60 mm versus 2 (unitless) mean with respect to CDF? Also please specify where these example data came from.

Thank you. The following sentence was added at the end of the paragraph associated with this figure: "At the $80^{th}$ quantile of the CDF, a precipitation value of $\sim 20$ mm in the non-transformed space represents a precipitation value of $\sim 1$ in the transformed space." The figure caption was also

changed to "Example of daily precipitation data normal score transformation for available gauges on 2019-06-20." to indicate this example is from our available gauges for that specific date.

Line 232: Please specify why the lapse rate relationships (and associated rate caps) in the Marmot Creek Research Basin have been selected here.

Thank you for the suggestion. Marmot Creek Research Basin has been selected because it is one of the most well-placed vertical profiles of snowfall gauges and it has a long precipitation lapse rate time series within our precipitation database. The following sentence "This vertical gauge profile was selected for its central placement and long precipitation lapse rate time series" was added at the end of this paragraph for clarification.

Line 280-281: Please list average change values associated with each area when mentioning, "Uncertainty fell in Montana" and "uncertainty rose in the upper Bow River basin", etc. It would also be helpful to read additional details on these areas – similar to the way Mount Robson was listed as the study domain's highest peak.

Thank you. This paragraph will be revised and include more statistics when mentioning uncertainty increases and decreases inside the study domain. We will also include more local information on why these uncertainty differences exist.

Like 345: Please define "very significant."

Thank you for the recommendation. The following sentences were changed to better clarify this reference's findings: "Brunet and Milbrandt (2023) have demonstrated that optimally designed networks usually favour the placement of new gauges in mountain regions of Alberta and British Columbia, Canada. Brunet and Milbrandt's study suggests that, in some cases, the placement of two to three gauges can have a very significant impact on reducing network uncertainty. For instance, improving network areal coverage from ~ 1600 to ~ 1300 km$^2$/gauge can decrease current network uncertainty close to an optimally designed network created from a blank slate in Alberta."

Line 411-414: Within the previous discussion section, suggest the authors point to identified limitations to the methods and/or provide a clear rationale. The series of steps taken within the methodology is well cited, but could be better defended – e.g., why ordinary and indicator kriging over other methods used in the introduction? The answer may be because these methods are frequently used – not just in past studies but in ongoing snowpack and hydrologic modeling research and operational applications. On that note, it would be impactful if the authors linked this approach to modeled or satellite-based precipitation products, which likely use some of these in-situ observations and cover these domains.

Thank you for this suggestion. We will include more details about the methods' limitations and a clearer rationale in the Results and Discussions section. We will also better describe the selection of techniques in the Material and Methods section. The choice of kriging techniques was due to their ability to provide an estimate of interpolation uncertainty. For comparison purposes, we will include a more direct link between our precipitation and uncertainty estimates with current modeled and satellite-based precipitation products.

**References**

Brunet, D. and Milbrandt, J. A.: Optimal Design of a Surface Precipitation Network in Canada, J. Hydrometeorol., 24, 727–742, https://doi.org/10.1175/JHM-D-22-0085.1, 2023.

Gasset, N., Fortin, V., Dimitrijevic, M., Carrera, M., Bilodeau, B., Muncaster, R., Gaborit, É., Roy, G., Pentcheva, N., Bulat, M., Wang, X., Pavlovic, R., Lespinas, F., Khedhaouiria, D., and Mai, J.: A 10 km North American precipitation and land-surface reanalysis based on the GEM atmospheric model, Hydrol. Earth Syst. Sci., 25, 4917–4945, https://doi.org/10.5194/hess-25-4917-2021, 2021.

Li, Y., Li, Z., Zhang, Z., Chen, L., Kurkute, S., Scaff, L., and Pan, X.: High-resolution regional climate modeling and projection over western Canada using a weather research forecasting model with a pseudo-global warming approach, Hydrol. Earth Syst. Sci., 23, 4635–4659, https://doi.org/10.5194/hess-23-4635-2019, 2019.

---

## Author Comment (AC2)

**Quantifying Spatiotemporal and Elevational Precipitation Gauge Network Uncertainty in the Canadian Rockies**

**Bertoncini and Pomeroy (2024)**

**HESS Manuscript EGUSPHERE-2024-288**

**Author Comment 2**

**\*Note that authors responses are in blue.**

**RC2 Comments**

Bertoncini and Pomeroy present and interesting approach for quantifying the uncertainty associated with sampling the spatiotemporal and elevational variability of precipitation in complex terrain, in addition to offering to the community a unique dataset for the Canadian Rockies.

General comments:

The paper is well written. It is relatively easy to understand the methods and follow the discussion, and the figures are well done. The literature review is interesting, but not exhaustive on the important topic of precipitation lapse-rate estimation methods. In particular, Dura et al. (2024, EGUsphere) proposes a few more relevant references. The dataset obtained is unique and helps to shed light on the important issue of network design in complex terrain. However, I must say that I am not totally convinced by some aspects of the methodology, partly because the authors do not evaluate the accuracy of the method through objective verification scores. This is important because the interpolation method proposed in the paper is new, even though it borrows from ideas already published in the literature to deal with elevational variability in precipitation intensity. In addition to the lack of verification, I also question some of the underlying hypotheses of the method itself. Overall, I believe that this paper has merit and should be published, but that important questions need to be answered first, and these will likely require additional experiments, as well as possible adjustments to the methodology.

Thank you for these comments. We will include a more extensive literature review on precipitation lapse-rate estimation methods and use Dura et al. (2024, EGUsphere) as a source of references. We will also evaluate the method's accuracy using objective verification scores. More details about how to implement an objective verification of the method and other required changes to the methodology will be described in the following responses.

In the next section, I propose specific comments for the introduction and methodology sections. I do not have much to say about the discussion part of the paper, although it might need to be revisited if changes are made to the methodology.

Concerning objective verification: verifying a gridded product that incorporates all available observations is obviously a challenge. I suggest that several test cases be chosen that correspond to various type of precipitation events. For these events, the interpolation procedure could be carried out using a subset of the available data, and the precipitation maps (with and without considering the altitudinal gradient in precipitation) could be compared to each other and to the precipitation observed at the remaining stations. Care should be taken to create a subset of data for verification that covers the distribution of elevations in the domain of interest. In addition to comparing kriging estimates to the observed precipitation, it is also necessary to compare the values of SD obtained from the kriging method to the distribution of the errors made when making predictions at unknown locations. In the NST transformed space, I would expect about two third of the estimations to fall within plus or minus one standard deviation from the kriging estimate. This should be checked. It is particularly important in this study because the discussion does not only focus on the estimate of precipitation, but also on its associated uncertainty.

Thank you for the suggestion. As you understand, carrying an objective verification of precipitation estimates that uses all available observations is challenging, so we have decided to perform a "leave-one-out" verification of the estimated precipitation. This technique involves estimating precipitation with the proposed technique removing one gauge at a time, which can be used to calculate objective verification scores. This procedure is repeated until all gauges are exhausted, and the objective verification scores are calculated for all the gauges in the dataset. We will also include an analysis comparing the kriging standard deviations with the error at unknown locations calculated with the "leave-one-out" technique. These analyses will be included in the next version of the manuscript.

Specific comments:

Line 60: "on the quantity variance". What do you mean by "quantity variance"? Not clear. Rephrase.

Thank you. The term "quantity variance" was used to describe the variance of any physical quantity, e.g., precipitation, temperature, relative humidity, etc. This sentence was modified to "Geostatistics techniques such as ordinary kriging (OK) can predict values in unobserved locations utilizing information on the variance of any physical quantity between a pair of station observations with a known distance." for better clarity.

Line 62: "...semivariogram, which is the relationship between the variance in the observed quantity with the measured distances." This is not true. The semivariogram is defined as $1/2\ E\ [\ (X_r - X_s)^2\ ]$ for random variables $X_r$ and $X_s$ at locations r and s. In its simplest expression, the semivariogram is only a function of the distance $d(r,s)$, but this is not the only option. The variance of $X_r$ and $X_s$ need not even exist for the semivariogram to exist, and in all cases the semivariogram does not measure the variance of the observed quantities $X_r$ and $X_s$ (well, actually it does if the distance is larger than the range of the variogram). You could write something like "semivariogram, which is

the relationship between the second moment of the differences between the observed quantity at two locations and the distance between these two locations".

Thank you, we agree with you. We modified this sentence to "which is the relationship between the second moment of the differences between the observed quantity at two locations and the distance between these two locations." to reflect your suggestion.

Lines 64-69: physically-based residual kriging, a.k.a optimal interpolation, should also be mentioned here. See for example Pelli et al. (2022, SERRA). In the Canadian context, Brasnett (1999, JAMC) used this approach to combine a background field from a numerical model with observations, taking into account elevation differences explicitly through a variogram, to interpolate snow depth observations.

Thank you for the suggestion. Optimal interpolation is an important interpolation technique that is related to our study. We have included the following sentence in this paragraph to address this technique: "Optimal interpolation (OI) is another method used to spatially estimate environmental variables while using physically based models as background for interpolation. OI has been used in a range of interpolation applications from groundwater information (Peli et al., 2022), snow depths (Brasnett, 1999), to precipitation in the Canadian Precipitation Analysis (CaPA) reanalysis product (Lespinas et al., 2015)."

Lines 81-82: be more specific here with respect to how meteorological models and hydrological models take into account lapse rates. These two categories of models do things very differently. Discuss briefly advantages/disadvantages of how it's currently done in these models. Add references.

Thank you. The following sentences were added in Line 82 to elaborate on the difference between how atmospheric and hydrological models consider precipitation lapse rates: "Atmospheric models simulate precipitation orographic enhancement by calculating moist air lifting and hydrometeor microphysics when passing over or near an orographic barrier (Houze, 2012; Lundquist et al., 2019). Hydrological models, on the other hand, employ observed or empirical lapse rates estimated from a profile of at least two gauges and distribute precipitation forcing based on the elevation difference between the precipitation source and the spatial modelling unit (Thornton et al., 1997; Liston and Elder, 2006; Smith and Barstad, 2004)."

Table 1: add references for the various instruments, either in the table or in the text.

Thank you. We will add references for the precipitation gauges utilized to compose the dataset.

Lines 159-160: I am not convinced that the use of ERA5-Land wind speed is acceptable for correcting precipitation accumulations for wind-induced undercatch. Have you compared ERA5-

Land wind speed predictions with observations in this region? ERA5-Land has a resolution of 9 km but relies on an atmospheric model which has a resolution of 30 km. The orography field used in numerical models is generally smoother than the nominal resolution, so the effective resolution of ERA5 is even lower than 30 km. Vanella et al. (2022, J.Hydrol.) have compared both ERA5 and ERA5-land wind speed predictions to observations in different climates and topography in Italy and concluded that both products strongly underestimate wind speed (by 28% to 42% depending on the region). Is it the same in your region? How does this affect your precipitation estimates?

Thank you for the suggestion. We understand this is a limitation to the methodology employed; however, the Canadian Rockies is a windy region, and wind undercatch in snowfall gauges is a major source of precipitation underestimation at some open sites. We note that many gauges (e.g., US SnoTel) are in small forest clearings where coarse-scale modelled wind speeds may overestimate wind speeds at gauge height in the clearing centre. We, therefore, decided to adopt the use of ERA5-Land for wind snowfall undercatch correction because even if there are known wind speed uncertainties in the order of 28 to 42%, as shown in Vanella et al. (2022), and unknown uncertainties due to local forest canopies, these uncertainties are not larger than for wind snowfall undercatch in the region. The latter uncertainty can decrease winter monthly precipitation amounts up to 72% in a high-elevation, unsheltered gauge (Pan et al., 2016). The following sentence was added to Line 164 to address this limitation: "The different techniques and meteorological data (observed versus ERA5-Land) to correct snowfall for wind undercatch may cause inconsistencies in the precipitation dataset since it is known that ERA5-Land wind speed can be underestimated by 28 to 42% (Vanella et al., 2022); however, some snow gauge sites are in forest clearings that are sheltered from the wind, and so these inconsistencies should be smaller than the impact of not correcting the dataset for wind undercatch. Wind snowfall undercatch underestimation in the region can be up to 72% of winter monthly amounts in a high-elevation, unsheltered gauge (Pan et al., 2016)."

Lines 162-164: I find it surprising that you decided to use the BC and COOP data as is because no existing equations exist to do the bias correction. I assume that the BC gauges are impacted by wind? I read that these gauges use a pressure transducer to estimate precipitation. They might also be impacted by changes in atmospheric pressure. Wrt ruler-based snowfall, how is the density of fresh snow estimated in order to obtain a water equivalent? Does it / should it take into account wind speed? If you cannot correct the data, it might be a better idea to reject the data when the wind speed is above a threshold such that biased estimates are expected.

Thank you for your comment. We understand that from a precipitation database composition perspective, one would want to keep only precipitation gauges that maintain the exact same standard of quality control. However, the goal of this manuscript is also to assess precipitation gauge network spatiotemporal and elevational uncertainty in the region for hydrological purposes. With that in mind, hydrologists make use of the available precipitation forcing in the region, while being aware of its uncertainty. Therefore, it is important to maintain gauges that employ different techniques such as from the BC and COOP networks which are still considered capable of reliable precipitation estimation in the region. The BC standpipes have been shown to have a

precipitation measurement precision ranging from 0.1 to 1 mm (Sha et al., 2021). They are always located in sheltered clearings where wind undercatch is minimized and considered small. We have corrected the description of COOP precipitation measurements. Snowfall (water equivalent) can be either calculated from a manual ruler and melting the amount of snow inside a sampler or using the US National Weather Service (NWS) 4 or 8-inch rain gauges and then melting the snow collected inside these gauges (NWS, 2013). COOP stations are known for having a daily negative observer bias of 1.27 mm and an observer tendency to round measurements to the nearest 0.1 or 0.05 inches (Daly et al., 2007); however, they are considered reliable stations that compose over 75% of daily precipitation stations in the US (Daly et al., 2021). In this region, they tend to be located in sheltered valley bottom sites where wind redistribution is minimal. Although an important topic, it is outside the scope of our study to develop wind undercatch corrections for standpipe, ruler-based, and NWS rain gauge snowfall measurements. The discussion above and additional references will be included in Section 2.2 to clarify the uncertainties in the networks utilized.

Line 193: the cubic-root transform is also used, see for example Lespinas et al. (2015) as well the more general Box-Cox transform, see van Hyfte et al. (2023, Tellus A).

Thank you for the suggestion. To make this background information more comprehensive, we added the suggested references as follows: "Although many transformations have been commonly applied in the past for implementation simplicity (e.g., log-normal, square-root, cubic-root, and Box-Cox) (e.g., Schuurmans et al., 2007; Sideris et al., 2014; Lespinas et al., 2015; van Hyfte et al., 2023),"

Line 205: how is the semivariogram model chosen? With what frequency is each model chosen?

We fit theoretical semivariograms (from the options gaussian, exponential, spherical, or penta-spherical) to the daily precipitation sample semivariogram using the *fit.variogram* function in the *gstat* R package. This fit is done using a least squares residual technique. An initial estimate of the sill, range, and nugget semivariogram parameters is made based on the shape of the sample semivariogram using the *autofitVariogram* function in the *automap* R package. The sentence in Line 206 was modified to reflect the above: "The choice of semivariogram model options was based on the most frequent models in Ly et al. (2011), which evaluated 30 years of best fitted daily semivariogram models, and the availability in R's *gstat* package. The semivariogram model was selected based on the smallest least squares residuals between theoretical and daily precipitation sample semivariograms using the *fit.variogram* function in the *gstat* R package. An initial estimate of the sill, range, and nugget semivariogram parameters were calculated based on the shape of the sample semivariogram using the *autofitVariogram* function in the *automap* R package." The frequency at which each model was selected will be included in the next version of this manuscript.

Lines 207-208: you mention that grid longitude and latitude is used to interpolate. When computing distances, does the method take into account the distortion caused by the fact that degrees of longitude are further apart in the south than in the north of the domain? Given the shape of your domain, this is important.

Thank you for the comment. All geospatial data is in WGS84, which is a geographic coordinate system that uses longitude and latitude coordinate values, thus not undergoing major differences in the distance represented by one degree of longitude in the south or north parts of the domain. The following sentence was added to the end of this paragraph for clarity: "All geospatial data is in the WGS84 geographic coordinate system; hence, not undergoing major differences in the distance represented by one degree of longitude in the south or north parts of the domain."

Line 208: back-transforming the data introduces a bias in the interpolation, because the kriging method is unbiased in the transformed space, but not in the original space. See van Hyfte et al. (2023) for details and formulas for taking this into account in the context of the Box-Cox transformation. Can you comment on the magnitude of that bias? Note that there is also a bias in the back-transformed standard deviation.

Thank you for the comment. We understand that back-transforming precipitation and standard deviation values to mm/day units has a bias associated with it. However, we assumed this bias to be negligible. The study by van Hyfte et al. (2023) showed that correcting for this type of bias in a Box-Cox transformation only slightly improves precipitation estimates during summer months. The Box-Cox transformation is similar to the NST transformation adopted here (Cecinati et al., 2017). The following sentence was added in Line 209 for further clarification: "Although there are known biases associated with back-transforming precipitation and standard deviation values, van Hyfte et al. (2023) has shown that correcting for this type of bias in a Box-Cox transformation only slightly improved precipitation estimates during summer months. The Box-Cox transformation is similar to the NST applied here (Cecinati et al., 2017)."

Line 216: you mention using 53 gauge pairs to estimate the lapse rate. I assume that you are recomputing the lapse rate on a daily basis? If so, does that number vary over time or are these 53 gauge pairs available for the whole time period covered by the product? Are these gauge pairs distributed relatively evenly over the domain? If not, do you think that this is problematic? Please provide a map of the location of these gauge pairs. Perhaps this information can be added to fig 1.

Thank you for the suggestion. Yes, the lapse rates are computed daily, as stated in Line 217. Yes, the number of gauge pairs also varies daily, which can be one reason that elevational uncertainty can increase, as stated in Lines 233-234. All the available same-slope elevational profiles of gauge pairs were considered within the domain. The number of pairs is relative to gauge density, i.e., more in the south and less in the north part of the domain, causing an underrepresentation of lapse rates in the north. We will add these pairs of gauges to Figure 1 and comment on their spatial representativeness in the next version of this manuscript.

Line 226-227: Because you have interpolated the raw data to obtain the final daily horizontal precipitation field, you now need to rely on a reference elevation field interpolated from gauge elevations. I assume that you are recomputing this reference elevation field every day, since the network changes over time. Can you please confirm? Note that an alternative approach would be to bring all station observations to a reference elevation using equation (3) for each station, e.g. 1500m and generate a precipitation field valid at that altitude. Then, a daily lapsed precipitation could be generated on the SRTM elevation model by applying again equation (3) for each SRTM grid cell using a reference elevation of 1500m for each grid point. It would be interesting to compare the two approaches.

Thank you for your comment. Yes, the reference elevation interpolated from gauge elevations is generated daily. Yes, we were aware that this could have been one of the paths we could have taken; however, it is outside the scope of this study to compare both methods. Therefore, we decided to follow the recommendations by Liston and Elder (2006) to be consistent with the literature.

Lines 233-234: this sentence does not read well. I do not fully understand what is meant here. In particular, the use of "Therefore" at the start of the paragraph suggests that what is being said follows logically from the end of the previous paragraph. However, the link is not obvious. Please rephrase. Perhaps there is a sentence missing just before this one?

Thank you for your comment. The "Therefore" was misplaced in the sentence. We have modified these two sentences at the beginning of this paragraph for clarity: "The reasoning for uncertainty estimation in this study is that uncertainty in interpolated lapsed precipitation fields is not only caused by uncertainty in spatial interpolation but also in precipitation lapse rate. Therefore, if fewer pairs of gauges exist at high elevations or the precipitation events happening on a particular day have diverging lapse rates, the spatiotemporal and elevational uncertainty is increased."

Lines 236-238: I believe that the term "coefficient of variation" and its abbreviation "CV" is an abuse of terminology. Indeed, if I understand well, the authors are computing the average of the SD field (over a year) and dividing by the average of the precipitation. However, the average daily value of SD does not correspond to the uncertainty associated with the mean annual precipitation. For example, if errors in daily precipitation estimates were independent, the standard deviation of the error on the annual average would correspond to the square root of the sum of the square of the SD values, divided by the square root of 365 days. Hence, if SD values were relatively constant over a year (they are obviously not), the standard deviation of the error on the annual average would be reduced by a factor of about twenty. The coefficient of variation of this error would be reduced by the same factor. Whatever the covariance structure of errors in daily precipitation estimates, the coefficient of variation would be obtained by manipulating the error variances (the square of SD), and not the standard deviation of the errors. However, the covariance between errors

in daily precipitation estimates has not been modelled in this study – it is thus unknown. For this reason, it is an abuse of terminology to talk about the term "coefficient of variation". What can be done? Well, the coefficient of variation can be computed for each day and each grid point, by dividing SD by the precipitation amount. Obviously, this is problematic for days without precipitation. It would still be possible to compute quantiles of the daily CV values and map one of them, for example a quantile halfway between the frequency of zeros and one. Another solution is simply to change the nomenclature and call the quantity reported in this paper something else than CV. This is what I suggest.

Thank you for your suggestion. The annual CV was actually calculated by dividing the annual accumulated standard deviation (i.e., the sum of the 365 days) by the annual accumulated precipitation, as stated in Lines 236-239. We have not worked with annual means in any of our calculations. Since the estimated annual spatiotemporal and elevational uncertainty represents the standard deviation in the CV equation (CV = SD/mean) and the estimated annual accumulated precipitation represents the mean, we believe the term CV is appropriate in this context. Please let us know whether you agree with us in light of this clarification.

**References**

Brasnett, B.: A global analysis of snow depth for numerical weather prediction, J. Appl. Meteorol., 38, 726–740, https://doi.org/10.1175/1520-0450(1999)038<0726:AGAOSD>2.0.CO;2, 1999.

Cecinati, F., Wani, O., and Rico-Ramirez, M. A.: Comparing Approaches to Deal With Non-Gaussianity of Rainfall Data in Kriging-Based Radar-Gauge Rainfall Merging, Water Resour. Res., 53, 8999–9018, https://doi.org/10.1002/2016WR020330, 2017.

Dura, V., Evin, G., Favre, A.-C., and Penot, D.: Spatial variability of seasonal precipitation lapse rates in complex topographical regions-application in France, 1–34, pre-print, 2024.

Daly, C., Gibson, W. P., Taylor, G. H., Doggett, M. K., and Smith, J. I.: Observer bias in daily precipitation measurements at United States Cooperative Network Stations, Bull. Am. Meteorol. Soc., 88, 899–912, https://doi.org/10.1175/BAMS-88-6-899, 2007.

Daly, C., Doggett, M. K., Smith, J. I., Olson, K. V., Halbleib, M. D., Dimcovic, Z., Keon, D., Loiselle, R. A., Steinberg, B., Ryan, A. D., Pancake, C. M., and Kaspar, E. M.: Challenges in Observation-Based Mapping of Daily Precipitation across the Conterminous United States, J. Atmos. Ocean. Technol., 38, 1979–1992, https://doi.org/10.1175/JTECH-D-21-0054.1, 2021.

Houze, R. A.: Orographic effects on precipitating clouds, Rev. Geophys., 50, 1–47, https://doi.org/10.1029/2011RG000365, 2012.

Lespinas, F., Fortin, V., Roy, G., Rasmussen, P., and Stadnyk, T.: Performance Evaluation of the Canadian Precipitation Analysis (CaPA), J. Hydrometeorol., 16, 2045–2064, https://doi.org/10.1175/JHM-D-14-0191.1, 2015.

Liston, G. E. and Elder, K.: A Meteorological Distribution System for High-Resolution Terrestrial Modeling (MicroMet), J. Hydrometeorol., 7, 217–234, https://doi.org/10.1175/JHM486.1, 2006.

Lundquist, J., Hughes, M., Gutmann, E., and Kapnick, S.: Our skill in modeling mountain rain and snow is bypassing the skill of our observational networks, Bull. Am. Meteorol. Soc., 100, 2473–2490, https://doi.org/10.1175/BAMS-D-19-0001.1, 2019.

Ly, S., Charles, C., and Degré, A.: Geostatistical interpolation of daily rainfall at catchment scale: The use of several variogram models in the Ourthe and Ambleve catchments, Belgium, Hydrol. Earth Syst. Sci., 15, 2259–2274, https://doi.org/10.5194/hess-15-2259-2011, 2011.

NWS: Snow Measurement Guidelines for National Weather Service Surface Observing Programs, 1–14 pp., 2013.

Pan, X., Yang, D., Li, Y., Barr, A., Helgason, W., Hayashi, M., Marsh, P., Pomeroy, J., and Janowicz, R. J.: Bias corrections of precipitation measurements across experimental sites in different ecoclimatic regions of western Canada, Cryosphere, 10, 2347–2360, https://doi.org/10.5194/tc-10-2347-2016, 2016.

Peli, R., Menafoglio, A., Cervino, M., Dovera, L., and Secchi, P.: Physics-based Residual Kriging for dynamically evolving functional random fields, Stoch. Environ. Res. Risk Assess., 36, 3063–3080, https://doi.org/10.1007/s00477-022-02180-8, 2022.

Schuurmans, J. M., Bierkens, M. F. P., Pebesma, E. J., and Uijlenhoet, R.: Automatic prediction of high-resolution daily rainfall fields for multiple extents: The potential of operational radar, J. Hydrometeorol., 8, 1204–1224, https://doi.org/10.1175/2007JHM792.1, 2007.

Sha, Y., Gagne, D. J., West, G., and Stull, R.: Deep-learning-based precipitation observation quality control, J. Atmos. Ocean. Technol., 38, 1075–1091, https://doi.org/10.1175/JTECH-D-20-0081.1, 2021.

Sideris, I. V., Gabella, M., Erdin, R., and Germann, U.: Real-time radar-rain-gauge merging using spatio-temporal co-kriging with external drift in the alpine terrain of Switzerland, Q. J. R. Meteorol. Soc., 140, 1097–1111, https://doi.org/10.1002/qj.2188, 2014.

Smith, R. B. and Barstad, I.: A Linear Theory of Orographic Precipitation, J. Atmos. Sci., 61, 1377–1391, https://doi.org/10.1175/1520-0469(2004)061<1377:ALTOOP>2.0.CO;2, 2004.

Thornton, P. E., Running, S. W., and White, M. A.: Generating surfaces of daily meteorological variables over large regions of complex terrain, J. Hydrol., 190, 214–251, https://doi.org/10.1016/S0022-1694(96)03128-9, 1997.

van Hyfte, S., Moigne, P. Le, Bazile, E., Verrelle, A., and Boone, A.: High-Resolution Reanalysis of Daily Precipitation using AROME Model Over France, Tellus, Ser. A Dyn. Meteorol. Oceanogr., 75, 27–49, https://doi.org/10.16993/tellusa.95, 2023.

Vanella, D., Longo-Minnolo, G., Belfiore, O. R., Ramírez-Cuesta, J. M., Pappalardo, S., Consoli, S., D'Urso, G., Chirico, G. B., Coppola, A., Comegna, A., Toscano, A., Quarta, R., Provenzano,

G., Ippolito, M., Castagna, A., and Gandolfi, C.: Comparing the use of ERA5 reanalysis dataset and ground-based agrometeorological data under different climates and topography in Italy, J. Hydrol. Reg. Stud., 42, 101182, https://doi.org/10.1016/j.ejrh.2022.101182, 2022.

---

## Author Response (AR2)

**Quantifying Spatiotemporal and Elevational Precipitation Gauge Network Uncertainty in the Canadian Rockies**

**Bertoncini and Pomeroy (2024)**

**HESS Manuscript EGUSPHERE-2024-288**

**\*Note that authors responses are in blue.**

**RC2 Comments**

I thank the authors for a very thorough response to my initial comments. I would like to re-emphasize one point: it is problematic to perform variographic analysis and interpolation in geographic coordinates (latitude and longitudes in degrees), because at mid-latitude, a degree of longitude is much shorter (in km) than a degree of latitude. Furthermore, the study domain is very elongated, covering around 8 degrees of latitude. Hence, the ratio of the length (in km) of a degree of longitude to the length of a degree of latitude varies significantly from the north to the south of the domain. Because, as I understand, the variographic analysis and interpolation were performed in geographic coordinates, I believe this choice has introduced an unwanted anisotropic effect on the variographic analysis (including potentially both on the choice of the variogram model and on the variogram parameter values), on the interpolated precipitation field and on the assessment of the uncertainty associated with the precipitation analysis. If it is indeed the case, the impact of this modelling choice on the paper results, findings and conclusions needs to be thoroughly assessed.

Thank you for the comment and suggestion. We have performed a similar leave-one-out validation presented in Section 3.2, but with a modification that ensures variographic analysis and kriging interpolation are performed using great-circle distances in kilometres – these distances should be the same no matter the latitude. The leave-one-out validation with distances in kilometres revealed a mean correlation of 0.52, bias of -1.00 mm/day, and RMSE of 4.96 mm/day for the 2020 year. These results present a slight degradation compared to the results computed in degrees for the year 2020 shown in Section 3.2; therefore, showing that in this case there was not a large difference between computing the estimates using geographic or projected (in kilometres) distances. However, the removal of one particular high-elevation gauge (Fisera Ridge) during the validation using kilometric distances made the interpolation less robust for the reference elevation surface $z_0^{i,j}$, which required that more variogram models were used as options. Additionally, to the Linear and Spherical models previously used, Penta-spherical, Gaussian, and Exponential models were also allowed. Still for 9 days of the 2020 year, $z_0^{i,j}$ could not be fitted and precipitation was set zero. We included the following paragraph to report these results and their implications in Lines 345-355: "Because geographic coordinates were used in this study to calculate the variograms and perform kriging interpolation, a separate leave-one-out validation using great-circle distances in kilometres for the 2020 WY was conducted to rule out

any major anisotropic effects that this choice could have introduced in the results. The leave-one-out validation using kilometric distances presented the following mean statistics: correlation = 0.52, bias = -1.00 mm/day, and RMSE = 4.96 mm/day. These statistics reveal a slight degradation of using distances in kilometres over using them in degrees. In addition, there were difficulties of fitting the reference elevation surface $z_0^{i,j}$ for one particular high-elevation gauge (Fisera Ridge at 2325 m), which required the use of additional variogram models. Still, $z_0^{i,j}$ was not fitted for nine days of the 2020 WY for this particular gauge and precipitation was set to zero. Therefore, it is concluded that using distances in degrees did not have any major influences on the precipitation estimates for the conditions of this study and the alternative approach introduced uncertainties into the analysis. Future studies should assess whether degrees or kilometric distances are the better choice for their domain conditions."